

# The changing composition of the Gulf of St. Lawrence inflow waters observed from transient tracer measurements

Lennart Gerke[1,5], Toste Tanhua[1], William A. Nesbitt[2], Samuel W. Stevens[3,4], Douglas W. R. Wallace[2]

[1]GEOMAR Helmholtz Centre for Ocean Research Kiel, Kiel, 24148, Germany
[2]Department of Oceanography, Dalhousie University, Halifax, B3H 4R2, Canada
[3]Department of Earth and Ocean Sciences, The University of British Columbia, Vancouver, V6T 1Z4, Canada
[4]Hakai Institute, Heriot Bay, British Columbia, V9W 0B7, Canada
[5][C]Worthy, LLC, Boulder, CO, 80302, USA

*Correspondence to*: Lennart Gerke (lennart@cworthy.org)





**Abstract.** The deep waters of the Gulf of St. Lawrence (GSL) have experienced a significant reduction in dissolved oxygen content during the past decades. One widely documented driver of this deoxygenation is a change in the composition of the

deep inflowing water that ventilates the Gulf. This deep water is known to consist of a mix of warmer, less-oxygenated North Atlantic Central Waters (NACW) and cooler, more-oxygenated Labrador Current Waters (LCW), with prior studies inferring a shift towards increased NACW contribution. However, this compositional change has only ever been inferred indirectly from physical and biogeochemical measurements via the use of inverse methods such as water mass analysis. In this study, we present results from the first spatially-comprehensive deep water transient tracer surveys in the GSL, allowing us to directly

map mean age estimates and use these to infer recent changes in the composition of regional deep waters. The results reveal an unexpected age distribution, with 'older' deep waters present near the Gulf's entrance, whereas 'younger' water is found further inshore, contrary to the expected estuarine circulation pattern, which transports deep water inland (increasing age along the flow path). This implies a gradual increase in the proportion of NACW from inshore areas towards the Gulf's entrance and provides direct evidence that the shift towards NACW dominated deep waters is ongoing as of 2022, contrary to earlier

predictions of the complete disappearance of the younger, well-oxygenated LCW.

## 1 Introduction

Ocean ventilation is a physical process that involves the dynamic exchange of properties like heat and dissolved gases, including oxygen, between the atmosphere and the ocean's interior (Azetsu-Scott et al., 2005; Fine, 2011; Talley et al., 2016).

It describes the transport of surface waters to the interior of the ocean, where the age of the water refers to the time since a water parcel was last in contact with the atmosphere. This process is, for example, important in distributing oxygen to deeper ocean layers and in shaping long-term properties of water masses in the interior, in contrast to the rapidly changing surface water layers. Ventilation studies often rely on measurements of transient tracers, such as sulfur hexafluoride ($SF_6$) and chlorofluorocarbon-12 (CFC-12), anthropogenic gases which are inert once dissolved in seawater and whose input can be

characterized by known variations of concentrations in the atmosphere (e.g. Khatiwala et al., 2001; Stöven et al., 2015). After the surface water is shielded from the atmosphere due to transport into the interior, the transient tracer concentrations are only affected by interior mixing processes since these compounds are stable in seawater during most conditions. Ventilation has been well-studied in certain regions of the world's oceans, but remains largely unexplored in others, particularly in marginal seas such as the Gulf of St. Lawrence (GSL).

Recently, the GSL region has garnered attention due to a significant decline in dissolved oxygen (DO) concentrations within the deep layer (e.g. Blais et al., 2024; Genovesi et al., 2011; Gilbert et al., 2005; Thibodeau et al., 2006), leaving a growing area persistently hypoxic (DO < 62.5 µmol L$^{-1}$ (micromoles of $O_2$ molecules present in 1 L of seawater)) (Jutras et al., 2023b). Among other property changes, it was reported that the annual average of DO at 300 m depth in the Estuary of the Gulf dropped to 37 µmol L$^{-1}$ in 2023 (12 % saturation) (Blais et al., 2024). This phenomenon has been observed increasingly in coastal zones





worldwide during the past few decades (Diaz and Rosenberg, 2008; Gilbert et al., 2005), for example due to benthic respiration (Lehmann et al., 2009), upwelling of low-oxygen waters (Grantham et al., 2004) and long residence times in ocean channels enhancing hypoxia (Fennel and Testa, 2019). In the well-studied region at the head of the Lower St. Lawrence Estuary (LSLE), deep layer DO values decreased from 130 µmol L$^{-1}$ to 60 µmol L$^{-1}$ between 1930 and 1984 where they then remained stable until about 2019. In 2020, concentrations rapidly decreased to an annual minimum of 35 µmol L$^{-1}$ (Jutras et al., 2023b), with

further, slight decrease observed in 2022 and 2023 by Blais et al. (2023, 2024), down to 34 µmol L$^{-1}$. Based on extensive measurements in this region, the hypoxic zone was first documented in the LSLE (Gilbert et al., 2005), with two-thirds of the decrease in DO since 1930 being estimated to be caused by a change in the composition of the deep water masses entering the GSL from the North Atlantic, and the other third being due to eutrophication (Gilbert et al., 2005; Mucci et al., 2011). The oceanic inflow supplies the deep water layers of the GSL with oxygen, and is known to consist of a mix of well-oxygenated

Labrador Current Water (LCW) and less-oxygenated North Atlantic Central Water (NACW). Several studies, such as Gilbert et al. (2005) and Jutras et al. (2020), hypothesised a shift in the composition of the deep water inflow toward a larger fraction of NACW using physical and chemical parameters, as well as inverse methods such as water mass analysis.

The study by Jutras et al. (2020) utilized the extended optimum multiparameter analysis (eOMP), which relies typically on conservative and quasi-conservative hydrographic properties such as temperature, salinity, nutrients, and/or oxygen. This study

incorporates transient tracer measurements (SF$_6$ and CFC-12) which provide additional insight into the ventilation timescales (mean age) of subsurface water parcels and enhances the analysis of water composition through the use of parameters that carry additional information about temporal changes and mixing processes.

This study first focusses on the independent use of transient tracers to estimate ventilation timescales within the deep water of the GSL via a transit time distribution (TTD) analysis. These tracer-derived mean ages are then integrated into a water mass

analysis to provide a detailed assessment of the water mass composition. This demonstrates that inclusion of tracer measurements improves the resolution of compositional shifts in the GSL and highlights that incorporating transient tracers is valuable to complement more traditional hydrographic data in methods such as eOMP.

## 2 Hydrography

The St. Lawrence Estuary and Gulf is one of the largest semi-enclosed estuaries in the world and consists of three components:

the Upper Estuary ranging from Ile d'Orleans to Tadoussac, the LSLE spanning from Tadoussac to Pointe-des-Monts, and finally the GSL. The GSL is a large marginal sea (approximate area of 240 000 km$^2$) located at the east coast of North America, and is connected to the Atlantic Ocean through Cabot Strait between Nova Scotia and Newfoundland in the southeast, and to the Newfoundland and Labrador Shelf via the Strait of Belle Isle in the northeast (see Figure 1). Due to its depth, Cabot Strait is the main entry point for North Atlantic water, which enters as a deep inflow (Koutitonsky and Budgen, 1991).



The GSL includes itself three channels (slightly darker blue areas in Figure 1), one south of Anticosti Island (Laurentian Channel), connecting the GSL to the Atlantic Ocean, one north of the Island (Anticosti Channel) and one channel towards the Strait of Belle Isle (Esquiman Channel), all bordering extensive shelf areas.

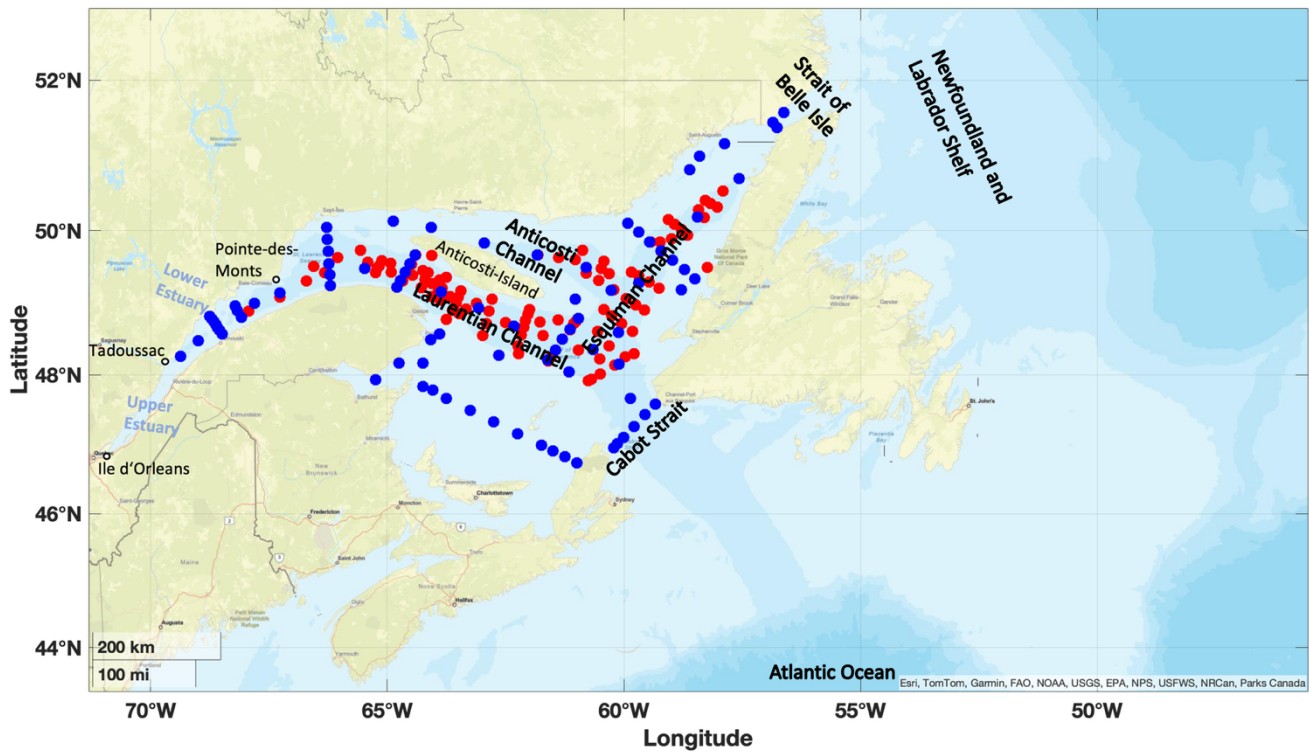

**Figure 1: The Gulf of St. Lawrence, with its main Channels and Straits identified, together with the Lower and Upper Estuary.**
**Sampling stations from the expeditions in June 2022 (TReX 2 – red dots) and November 2022 (DFO's AZMP survey/TReX 4 – blue dots) are displayed.**

The current that feeds deep water into the Gulf of St. Lawrence from the Atlantic Ocean carries two constituents, LCW and NACW, which enter at depths exceeding 200 m through Cabot Strait, forming the relatively warm and saline deep water of
the GSL ($\theta$ = 1 °C to > 7 °C and $S_p$ = 32.5 to 35) (Galbraith et al., 2024; Lauzier and Trites, 1958; McLellan, 1957).
In addition to the deep water entering the Gulf through Cabot Strait, the water column structure consists of an intermediate layer and a surface layer. The cold (< 1 °C) and saline intermediate layer (CIL – Cold Intermediate Layer), is mainly formed locally during winter due to interior water mixing when cooling and formation of sea ice increases the density of the surface water (Galbraith, 2006). A small fraction of the CIL also originates from the inflow of cold water from the Labrador Shelf
through the Strait of Belle Isle and an even smaller contribution originates through Cabot Strait at intermediate depths (Galbraith, 2006; Shaw and Galbraith, 2023). The intermediate water layer in the LSLE and Upper St. Lawrence Estuary consists of less salty water, which is renewed by saltier CIL water moving inland from the Gulf in spring (Galbraith, 2006).



The relatively warm in summer (> 1 °C) and low salinity surface water flows mainly seaward, thereby transporting the continental runoff freshwater from the Estuary, and water from the north shore rivers towards the Atlantic Ocean (Galbraith, 2006; Gilbert and Pettigrew, 1997). This outflow leaves the Gulf through the southern edge of Cabot Strait, as the inflow mainly occurs around the northern boundary of the strait (Galbraith et al., 2024).

Using ocean models, the transit times of the deep water within the Laurentian Channel transported from Cabot Strait until the head of the channel range from 2 to 7 years, depending on the depth (Saucier et al., 2003). For instance, Gilbert (2004) estimated water at depths of 250 m to flow inland from the Atlantic Ocean with a speed of 1 cm/s, reaching Cabot Strait after 1 year and the Estuary after 3.5 years. This results in a transit time of 2.5 years from Cabot Strait to the Estuary. Recently, Rousseau et al. (2025) provided and integrated estimates for the transit time of the entire deep water column (>225 m), being $3.2 \pm 0.7$ years from Cabot Strait to the head of the Laurentian Channel. The transit time was analyzed in detail by Stevens et al. (2024) including measurements of a released tracer, that estimated the time for the first water parcels to reach the head of the Gulf to be 1.7 years from Cabot Strait with the bulk of the water arriving after around 4.7 years (in approximate agreement with the earlier estimates of Bugden (1988)). This estimate represents waters centred around the density surface of $\sigma_\Theta = 27.26$ kg/m$^3$, representing the water layer from $\approx 250 - 310$ m within the Laurentian Channel.

As the transient tracer measurements that we present in this study were collected in tandem with data collected for the deliberately-released tracer experiment, as detailed in Stevens et al. (2024), we use the advection timescales presented in that study to inform our analyses. Furthermore, we also choose the same density surface of $\sigma_\Theta = 27.26$ kg/m$^3$ to represent the core of the deep water inflow.

## 3 Data and Methods

### 3.1 Surveys

The data analyzed in this study was obtained during two field campaigns, one in June 2022 (TReX 2) and one in November 2022 (DFO's AZMP Survey/TReX 4), both carried out on the Canadian research vessel '*R/V Coriolis II*'. The primary objective of the TReX 2 field campaign was to track the deliberately-released tracer trifluoromethyl sulfur pentafluoride (SF$_5$CF$_3$), which was release in October of 2021 on the density surface of $\sigma_\Theta = 27.26$ kg/m$^3$ within the Laurentian Channel. This release was part of the deep water tracer release experiment aiming at monitoring the transport pathways for the deep water in the GSL (TReX Deep; Stevens et al. (2024)). Since the released tracer and the transient tracers focused on in this study, CFC-12 and SF$_6$, were measured simultaneously, sampling was concentrated primarily on this density layer, resulting in an uneven depth-distribution of transient tracer observations.

The June cruise was supported by the Marine Environmental Observation, Prediction and Response Network (MEOPAR) and Réseau Québec Maritime (RQM), whereas the November cruise was conducted by the Department of Fisheries and Oceans, Canada (DFO) as part of the Atlantic Zone Monitoring Program (AZMP). The AZMP leads regular surveys along fixed





sections since 1999 to monitor physical and biological processes in the St. Lawrence Seaway and the coastal shelf of Eastern Canada, around Nova Scotia and Newfoundland.

Both cruises sampled sections in the Laurentian Channel, the Anticosti Channel, the Esquiman Channel, and the later cruise included sampled stations at Cabot Strait (see Figure 1) (Blais et al., 2023).

## 3.2 Observation parameters

The observed tracer concentrations of CFC-12 and $SF_6$ from the two cruises in 2022 were used to estimate water mass mean ages in the LSLE and GSL. Thanks to their time-varying historical concentration in the atmosphere and conservative behaviour in seawater, these transient tracers are useful to help quantify water mass ventilation time-scales. Assuming a saturation state that reflects equilibrium between surface water and atmosphere, knowing the solubility, and the tracers' atmospheric

concentration as a function of time (Bullister et al., 2002; Bullister and Warner, 2017; Warner and Weiss, 1985), it is possible to reconstruct the historical input functions of each tracer at the sea surface. Since the concentration of $SF_6$ continues to increase in the atmosphere, it can be used to estimate ventilation over approximately the last 40 years. CFC-12, on the other hand, is more appropriate for slightly older water masses, as its production started around 1940, but its atmospheric concentration has decreased since 2002 (see Figure S1). Nevertheless, measurements of more than one tracer are needed to provide information

on ventilation patterns and determine, for example, if this process is more advectively or diffusively dominated (e.g. Stöven et al., 2015).

The tracers were measured simultaneously on board of the research vessel with a gas chromatographic – electron capture detector system attached to a custom purge and trap unit (GC-ECD/PT5) (Bullister et al., 2002; Gerke et al., 2024; Tanhua et al., 2004, 2005b). The system has a detection limit of approximately 0.03 fmol/kg and 0.02 pmol/kg for $SF_6$ and CFC-12,

respectively (Stöven et al., 2015), and all measured concentrations during the two surveys showed values well above the system's detection limits. For details on the GC-ECD/PT5 system and data calibrations, see Appendix A.

The TTD (transit time distribution) method is a well-established and widely used concept to represent ventilation and allows calculation of various characteristics of the age distribution for a parcel of water in the ocean, including its mean age (e.g. Hall and Plumb, 1994; Shao et al., 2016; Trossman et al., 2012). As described in detail in Appendix B, we constrained the most

suitable TTD for this region using our measured tracers' concentrations in combination with a time dependent saturation estimated by Raimondi et al. (2021) and a $\Delta/\Gamma$-ratio (width of the distribution to mean age ratio) of 1.2. As only relatively young water is present within the Gulf, we calculated the final mean ages using the measured $SF_6$ concentrations in combination with the TTD which was determined through the use of both tracers. Note that in this study the term "age" refers to the time elapsed since a water mass was last in contact with the atmosphere, i.e. since it was ventilated, not the time since it

entered the GSL.



We use potential temperature (θ) and practical salinity ($S$p) to infer the composition of the Gulf inflow waters, and the transient tracers to analyze the ventilation timescale of these waters. The θ and $S_p$ data was obtained via CTD sensors during both surveys (e.g. Blais et al., 2023). In addition, both cruises included oxygen probe measurements (SVE-43 DO probe) calibrated by Winkler titration from collected water samples (Hansen, 1999).

Uncertainties in the mean age values arise from various factors, including uncertainties in the tracer measurements (≈ 2 %), uncertainties in constraining the TTD, and most significantly from the input functions used (i.e. atmospheric concentration and saturation assumption). Considering all these factors, an uncertainty of 10 % is applied to the calculated mean ages, represented by error bars. See Appendix C for details on each of the uncertainty factors.

To obtain information on transient tracer concentrations within the individual water masses – LCW and NACW – before
entering the GSL, we used data from the GLODAPv2.2022 (Global Ocean Data Analysis Project version 2) data product (Lauvset and et al., 2022). We focussed specifically on data collected after 2010 in the region near the mouth of the Laurentian Channel, where the deep water enters, while still representing LCW and NACW as distinct unmixed water masses. Mean ages were primarily computed from CFC-12 measurements due to higher abundance of this tracer in that region. The same saturation and Δ/Γ-ratio, as determined earlier were considered for the calculation.


### 3.3 Water mass analysis

To obtain detailed information on the water mass composition of the deep water, we use a linear fraction model including the three parameters, mean age (Γ), θ and $S$p (see Equations 1- 4) and solve it for a least square solution of fractions.

$$\Gamma_{obs} = f_{LCW} * \Gamma_{LCW} + f_{NACW} * \Gamma_{NACW} \tag{1}$$

$$\theta_{obs} = f_{LCW} * \theta_{LCW} + f_{NACW} * \theta_{NACW} \tag{2}$$

$$Sp_{obs} = f_{LCW} * Sp_{LCW} + f_{NACW} * Sp_{NACW} \tag{3}$$

$$1 = f_{LCW} + f_{NACW} \tag{4}$$

Here, the observed values represent the measured parameters within the GSL ($\Gamma_{obs}$; $\theta_{obs}$; $Sp_{obs}$) during 2022, $f_{LCW}$ and $f_{NACW}$
display the individual fractions of the two mixing water masses and $\Gamma_{LCW}$, $\Gamma_{NACW}$, $\theta_{LCW}$, $\theta_{NACW}$, $Sp_{LCW}$ and $Sp_{NACW}$ indicate the endmembers of the two water masses before mixing. The endmembers for temperature and salinity are set to be the mean of the θ and $S$p range of LCW and NACW, respectively, as presented in Jutras et al. (2020). For the mean age endmembers, we collated tracer data measured outside the GSL since 2010 in the respective temperature, salinity and density ranges, representing the LCW and NACW and computed the endmembers for the water mass analysis from the mean of the calculated
mean ages within the respective water mass (see Table 1). All parameters are considered quasi-conservative (i.e., not affected by biogeochemical processes) given the short transit time from immediately outside the GSL to the Laurentian Channel, making the use of mean age endmembers plausible (Tanhua et al., 2005a).





**Table 1: Endmembers of θ, *S*p and Γ used for the water mass analysis alongside each water mass fraction range. Note that the mean age endmembers are represented by the mean of all computed values, not representing the middle of the range.**

| Variables | NACW range | LCW range | NACW endmember | LCW endmember |
|-----------|------------|-----------|----------------|---------------|
| Θ [°C] | 4.4 – 8 | -0.7 – 3.2 | 6.2 ± 1.04 | 1.25 ± 1.13 |
| *S*p [psu] | 35 – 35.2 | 33.4 – 35 | 35.1 ± 0.06 | 34.2 ± 0.46 |
| Γ [years] | 49 – 105 | 7 – 19 | 86.5 ± 1.67 | 12.5 ± 0.23 |

This linear approach bears uncertainties, such as the selection of the individual endmembers, uncertainty in the parameter observations used for the fraction calculation, and structural model assumptions, including the equal weighting of parameters in a simple least square solution.

To address these uncertainties, we estimated the ambiguity in the selection of the mean age endmembers as the standard error of the mean (SEM) from the individual measurements within each water mass in the Atlantic Ocean. This resulted in endmember uncertainties of 1.67 years for NACW and 0.23 years for LCW. For temperature and salinity endmembers, uncertainties were approximated using uniform distributions across the reported parameter ranges, resulting in uncertainties of 1.04 °C and 1.13 °C for NACW and LCW temperatures, respectively, and 0.06 and 0.46 psu for NACW and LCW salinities.

The mean age observations within the Gulf, included in the fraction analysis, were assigned a relative uncertainty of 10 %, as previously specified, while errors in temperature and salinity observations were considered negligible. An additional 20 % relative uncertainty was introduced to account for structural assumptions in the simple linear mixing model. All uncertainties were jointly propagated using a Monte Carlo approach (N = 10,000) (JCGM, 2008), yielding in an overall LCW fraction uncertainty of approximately 21 %. A detailed uncertainty budget analysis is provided in Appendix D.

Despite these uncertainties, this method offers a comparison to the ventilation timescales derived from the transient tracer measurements and provides additional insights into water mass composition by more effectively capturing temporal changes and mixing processes in comparison to the other methods solely relying on hydrographic parameters.

### 3.4 Deep water time proxy at Cabot Strait

A proxy time series at Cabot Strait was constructed to better resolve temporal changes in DO and potentially related water mass properties, such as mean age or water mass composition. This approach provides a clearer view of long-term trends compared to basin-wide observations based on single-year measurements. To construct this deep water proxy time series, transient tracer and oxygen data from the two surveys collected throughout the Laurentian Channel were combined with the estimated deep water transit time from Stevens et al. (2024). A fixed location within Cabot Strait (47.2 °N; 59.7 °W) was selected, and the distance from each sampling point to this location was calculated. Using a transit speed of 0.5 cm/s, the time



at which each water parcel would have passed Cabot Strait was then computed, providing an estimate of the age at the Gulf's entrance. To account for changes in DO concentrations, the oxygen utilization rate (OUR) within the Laurentian Channel, estimated to be 21.1 μmol/kg per year (Nesbitt et al., 2025 - in revision), was considered in the calculation.

## 4 Results

### 4.1 Water mass distribution

During both cruises, all three water layers of the GSL, the warm and fresh surface, the cold intermediate and deep water, defined as in Galbraith et al. (2024), were clearly present (see Figure 2).

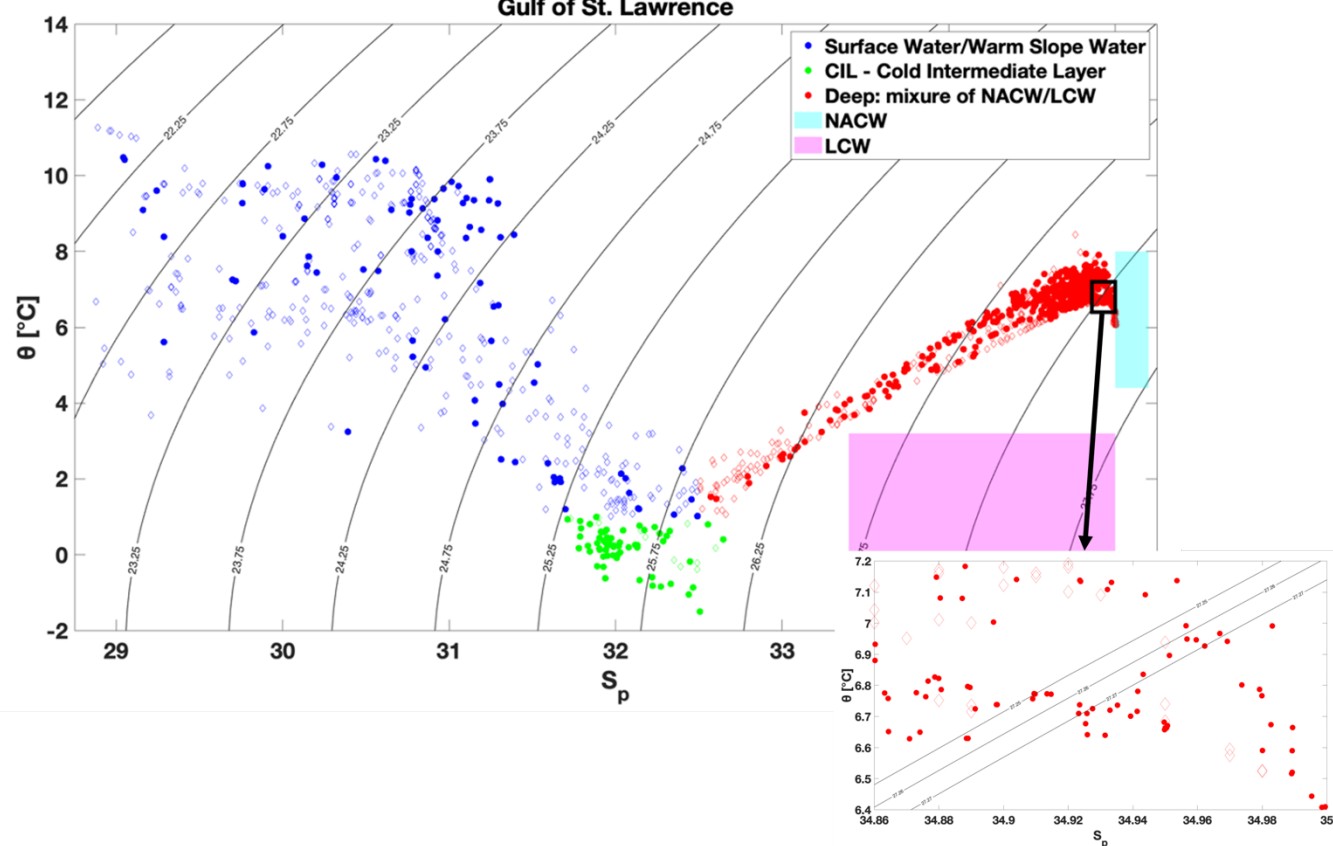

**Figure 2: Potential Temperature (θ) vs Practical Salinity ($S_p$) plot from the bottle data of the two cruises in 2022, showing the three main layers of the GSL, as indicated by different colors with density lines of $\sigma_\Theta$ (TReX 2 – dots; DFO's AZMP survey/TReX 4 – diamonds). The pink and the cyan areas represent the θ vs $S_p$ ranges of LCW and NACW, as defined in** Jutras et al. (2020). **In addition, a zoomed in area around the density of $\sigma_\Theta$ = 27.26 kg/m³ is shown, with density lines at 27.25, 27.26 and 27.27.**



The temperature and salinity structure throughout the water column was similar during both cruises (see Figure 2 – dots and diamonds) with the deep temperature maximum (8 °C) present at depths between 200 and 300 m in the Laurentian Channel: in recent years, the temperature at these depths has increased from an average of 5.2 °C in 2009 to 7 °C in 2022, with the warmest water being observed near Cabot Strait, where temperatures exceeding 7 °C were measured as early as 2012 (Galbraith et al., 2024). As our study and the sampling strategy of TReX 2 focus on the region's deep layer, we concentrate on properties

in the core of the deep water layer on the $\sigma_\Theta = 27.26$ kg/m$^3$ isopycnal (250 – 310 m) (see Figure 3). Note that on this isopycnal, temperatures vary from approximately 6.4 °C to 7.5 °C and salinity varies from 34.7 to 34.95 (see Figure 3c – 3f).

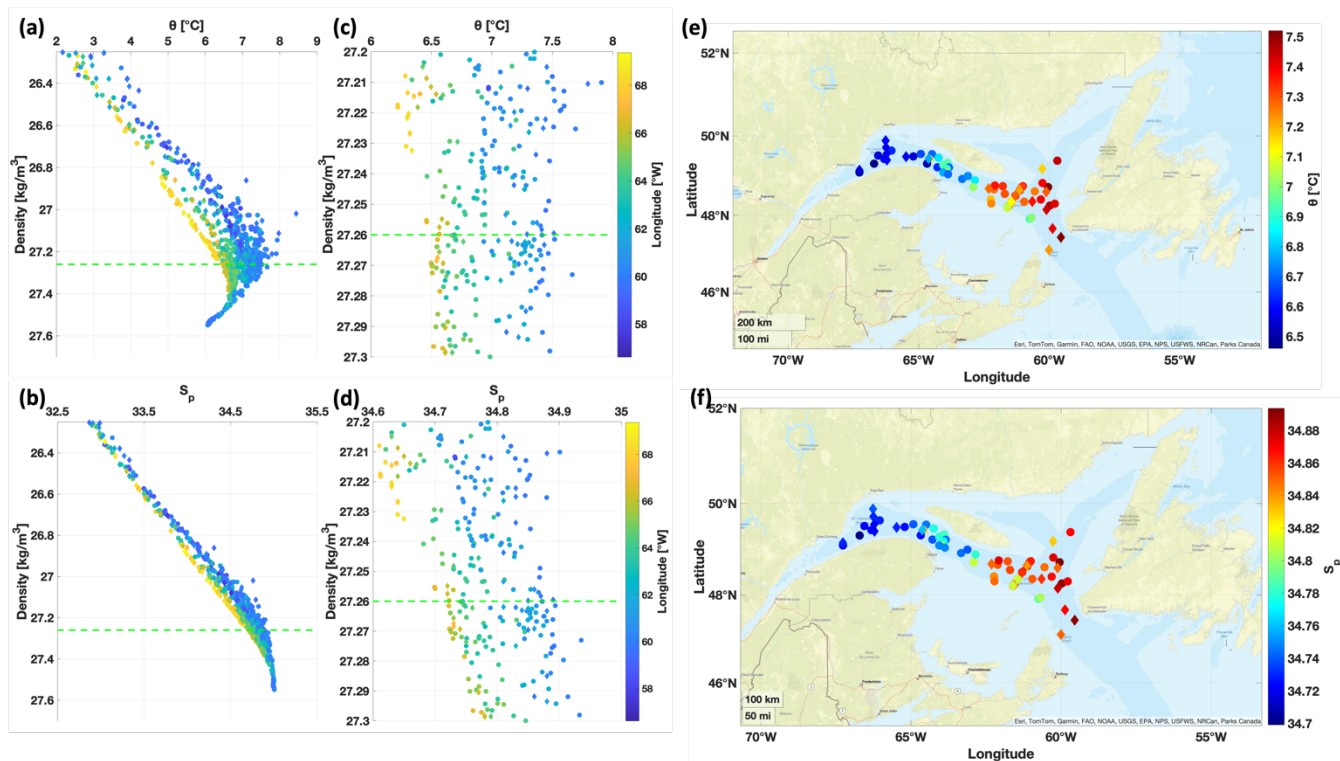

**Figure 3: Potential Temperature (a) and Practical Salinity (b) measured within the deep water layer in the GSL (density > 26.25**
**kg/m$^3$), with the colorbar representing the longitude. Figures c and d represent a zoomed in area around the $\sigma_\Theta = 27.26$ kg/m$^3$ isopycnal (green dotted line) and plots e and f show these measurements plotted on a map of the GSL against longitude and latitude. The dots represent the measurements from the earlier survey (TReX 2) and the diamonds the measurements from the later cruise (DFO's AZMP survey/TReX 4).**

Colder and less saline waters were observed to the west of the region within the entire deep water layer (see Figure 3a & 3b). Focusing on the $\sigma_\Theta$=27.26 kg/m$^3$ isopycnal (see Figure 3c – 3f) this is also evident, alongside a sudden discontinuity in both variables at approx. 63 °W (eastern tip of Anticosti Island).



## 4.2 Age distribution

When examining the mean age of water from tracer data collected since 2010 in the Atlantic region outside the GSL, focussing on a density of $\sigma_\Theta = 27.26$ kg/m$^3$ (see Figure 4b), we observe oldest water (70 – 105 years) to be located south and west of Cabot Strait with considerably younger water located to the northeast (5 – 20 years). This reflects the distribution of more recently ventilated LCW dominant in the northeast of the region and the older NACW in the south and southwest. The influence of the younger LCW is visible near the entrance to the Gulf, where there is a reduction in age of the northward-flowing NACW.

At the mouth of the Laurentian Channel, the age of the deep waters is around 40 – 50 years.

The observations of the GSL deep water shows that the water close to Cabot Strait is older compared to waters located closer to the Lower Estuary (see Figure 4a). This age distribution is consistent with distributions of temperature and salinity (see Figure 3), where 'younger' waters have higher LCW composition (colder and less salty, more oxygen) and 'older' waters have higher NACW composition (warmer and saline, less oxygen). Consistent with the temperature and salinity spatial variability,

deep layer mean age shows an abrupt 5 – 10 year shift towards younger waters at approximately 63 °W (i.e., the southeastern tip of Anticosti Island; see Figure 4c). On the $\sigma_\Theta = 27.26$ kg/m$^3$ isopycnal, we see a shift in age from around 45 years to 35 years (see Figure 4d). Apart from this sudden shift, the mean age generally increases gradually between the LSLE and Cabot Strait. On the studied isopycnal, mean age values are largest at Cabot Strait (60 years), with younger waters (40 – 50 years) present in the eastern part of the Laurentian Channel between 61 °W and 63 °W and the youngest waters located to the west

of 63 °W (30 – 40 years; see Figure 4a).

Even though we focused sampling on the $\sigma_\Theta = 27.26$ kg/m$^3$ isopycnal, samples collected on other isopycnals in the deep layer show comparable results, with younger water observed further inland relative to measurements taken closer to Cabot Strait (see Figure 4c). Additionally, in the intermediate layer, at densities between $\sigma_\Theta = 25.25 – 26.25$ kg/m$^3$, similar trends were observed in age and tracer concatenations despite the sparse data coverage.










**Figure 4: a) Mean age on the σ_Θ = 27.26 kg/m³ isopycnal in the Laurentian Channel plotted on a map. b) Mean ages interpolated to the density of σ_Θ=27.26 kg/m³ outside the GSL from measurements after 2010. c) SF₆ partial pressure and mean ages in the CIL and the entire deep layer of the Laurentian Channel, with d) zoomed in on around the σ_Θ = 27.26 kg/m³ isopycnal. The red dots display**
**samples measured further east and the blue dots the data from stations westward, with errorbars representing the concentration uncertainty of 2 % (pSF₆ plot), as well as the calculated mean age uncertainty of 10 %. The green line represents the isopycnal of σ_Θ = 27.26 kg/m³ and the red dotted lines the isopycnals of σ_Θ = 25.25 kg/m³ and σ_Θ = 26.25 kg/m³ marking the different water layers.**

### 4.3 Water mass analysis

When only considering θ and $S_p$ in the linear water mass fraction analysis, no presence of LCW on the σ_Θ = 27.26 kg/m³ isopycnal within the Laurentian Channel of the GSL was visible (see Figure S2). This occurs because both parameters, temperature and salinity, remain close to the range characteristics of the NACW endmember (see Figure 2), causing the method to attribute all water mass fractions exclusively to NACW. Although the analysis of the individual hydrographic parameters shows along channel variability and indicates the influence of the fresher LCW (see Figure 3), the least-square solution does

not capture this.

When considering a third parameter, mean age, in the water mass analysis, which differs significantly between the two source waters, the influence of LCW becomes more pronounced, particularly further inshore (see Figure 5). Despite inherent uncertainties that limit the interpretation of the estimated fraction values, observational evidence suggests that the fraction of LCW gradually decreases from the west to the east within the Laurentian Channel.



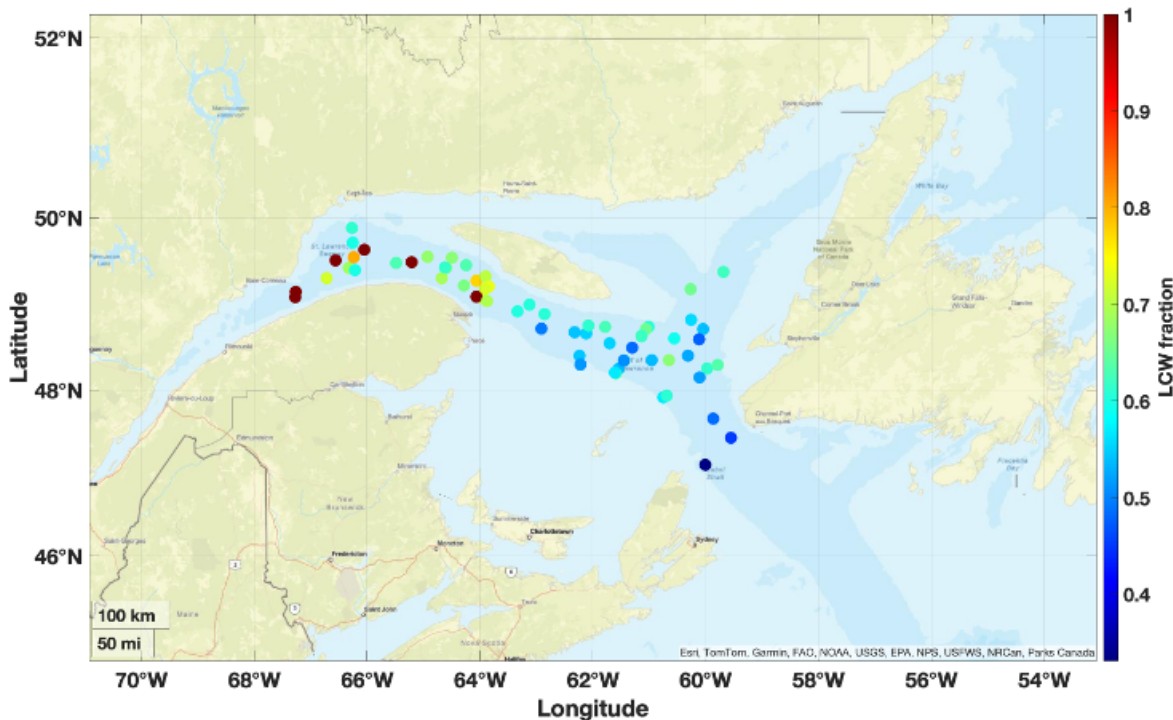


**Figure 5: LCW fraction on the $\sigma_\Theta$ = 27.26 kg/m³ isopycnal deep water in the Laurentian Channel from θ, $S_p$ and mean age (Γ) observations plotted on a map. The endmembers used in this calculation are provided in Table 1.**

## 4.4 Proxy Time series at Cabot Strait

The analysis of a proxy time series for mean age, LCW fraction and DO at Cabot Strait from 2018 to 2022 reveals an increase in mean age and decreased in the LCW fraction over time (see Figure 6 – upper and lower panel). Despite these changes, the DO concentrations remained relatively stable (middle panel), showing only a slight decrease over the time stretch, although with considerable variability.





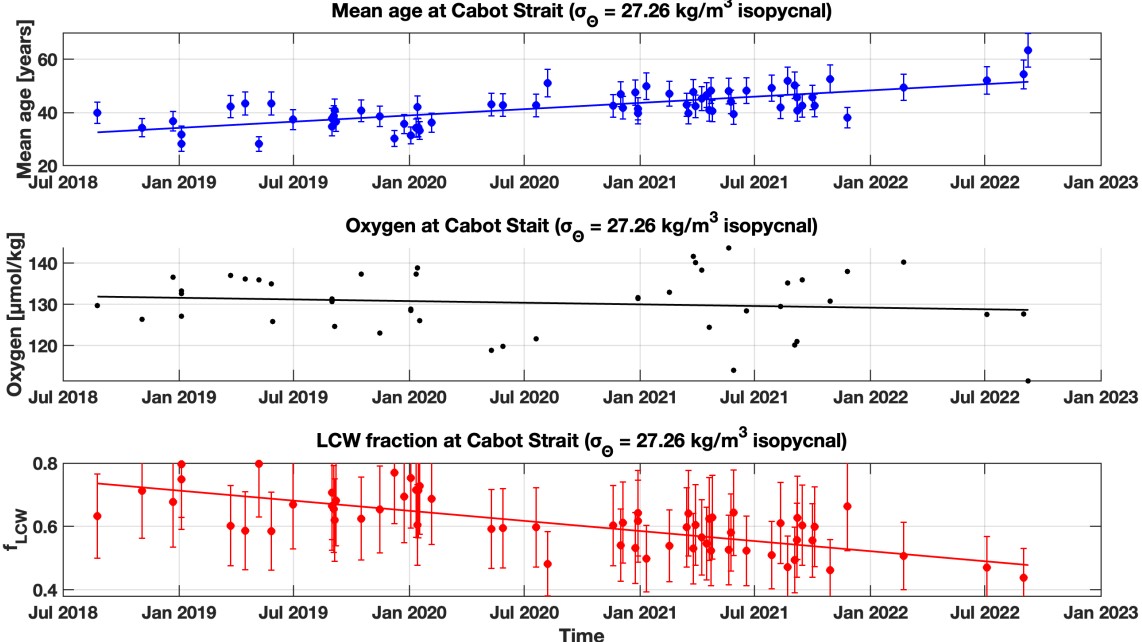

**Figure 6: A proxy timeseries of mean age (top), dissolved oxygen (middle) and LCW fraction (bottom) at Cabot Strait from 2018 to 2022. The error bars represent the 10 % uncertainty in the mean age calculation in the upper panel and the 21 % uncertainty from the water mass fraction uncertainty budget in the lower panel.**

## 5 Discussion

The temperature, salinity and mean age analysis of the deep water on the $\sigma_{\Theta} = 27.26$ kg/m³ isopycnal along the Laurentian channel show young, cooler, and fresher water in the Lower Estuary and western Gulf, with older, warmer and more saline water entering through Cabot Strait and within the eastern Laurentian Channel (Figures 3 and 4a). These are the opposite pattern of water mass age that might be expected if water is transported along the Laurentian Channel, westwards, from the Atlantic inflow region in the east (Dickie and Trites, 1983). These along-channel property gradients can be caused by vertical mixing within the Gulf, e.g. halocline vertical downward motion at intermediate water layers (Galbraith et al., 2024), and/or by changes in the water mass composition of the inflowing deep water. Apart from a rapid shift in all parameters around 63 °W towards older, warmer, and saltier water in the east, the mean age data shows a gradual age increase from the west to the east (see Figure 4a), also demonstrated in the proxy time series at Cabot Strait from 2018 to 2022 (see Figure 6 – upper panel).

These observations of water mass ages support the general hypothesis of a recent change in the composition of the Gulf of St. Lawrence's deep inflow towards an increased NACW composition (Galbraith et al., 2024; Jutras et al., 2020), characterized by older, warmer and saltier water. However, recent studies, such as Jutras et al. (2023b), suggest that since 2021 the deep water within the Laurentian Channel has been almost entirely composed of NACW. This finding contrasts with the gradual



west to east age increase along the Laurentian Channel observed in this study, suggesting that the influence of NACW on the deep water composition still continues to increase over time, as of 2022.

Supporting this interpretation, our water mass fraction analysis, incorporating mean age as a tracer with distinctly different source water signatures, indicates higher LCW fractions near the LSLE, gradually decreasing toward the eastern Gulf (see Figure 5). This implies that, as of 2022, NACW influence is continuing to rise throughout the Laurentian Channel on the $\sigma_\Theta = 27.26$ kg/m$^3$ isopycnal from the LSLE to Cabot Strait. Expressed differently, the significant differences in transient tracer concentrations and computed mean ages of the LCW and NACW, provide valuable insight into the evolving water mass

composition. While the calculated fraction values itself are subject to significant uncertainty due to model assumptions and endmember variability, the ongoing trend of increasing NACW influence remains identifiable.

A comparison of the computed fractions with values reported in previous studies reveals substantial variability. For example, Gilbert et al. (2005) described a shift from 72 % LCW and 28 % NACW in 1930 to 53 % LCW and 47 % NACW over the period of 1980 to 2003, consistent with estimates from Bugden (1988) when using their definition of LCW, which incorporates

temperature, salinity and oxygen characteristics. These values are similar to our observed fractions of 50 % at Cabot Strait in early 2021. In a more recent study, Jutras et al. (2020) reported LCW fractions of less than 10 % on the $\sigma_\Theta = 27.3$ kg/m$^3$ isopycnal in 2017, and around 50 % at $\sigma_\Theta = 27.5$ kg/m$^3$, indicating a stronger LCW influence at greater depths. In comparison, our analysis on the $\sigma_\Theta = 27.26$ kg/m$^3$ isopycnal, shows values of up to 60 % at Cabot Strait in 2018. In Jutras et al. (2023b), LCW fractions at Cabot Strait were reported as 0 % in 2021, with only small LCW influence at intermediate depths (~250 m)

between the middle of Anticosti Island to the LSLE. Interestingly, at greater depths, they found no LCW influence throughout the Laurentian Channel, in contrast to 2017, when higher LCW fractions were observed closer to the bottom (Jutras et al., 2020).

These recent studies primarily estimate water mass fractions within the Laurentian Channel deep water using basic hydrographic parameters (temperature, salinity, nutrients, oxygen, etc.) combined, for example, with an eOMP analysis,

comparable to our simplified water mass analysis approach using temperature and salinity (see Figure S2). The limitation of relying on, e.g., oxygen as an indicator for water mass analysis are evident in our proxy timeseries at Cabot Strait (see Figure 6 – middle panel), where only minor overall changes are detected despite high variability. Moreover, the individual θ and $S$p trends suggest ongoing changes (see Figures 3) and the θ vs. $S$p plot (see Figure 2) shows no overlap with the LCW range, indicating that analysis using only two variables may provide an incomplete picture (Liu and Tanhua, 2021).

Therefore, expanding this analysis to include tracer-derived mean ages, a parameter that reflects distinct different source water signatures, provides complementary insights and can help refine estimates of compositional shifts, offering a more accurate depiction of temporal changes and mixing processes. However, it is important to interpret these results with caution, given uncertainties of up to 21 % in our analysis, as well as the observed intensification of the eastward retroflection of the Labrador Current since 2016 (Jutras et al., 2023a), which has likely contributed to lower LCW fractions than those estimated here.

The oldest water with the lowest LCW fraction in 2022 is observed all the way near Cabot Strait, indicating that fractions of LCW were present throughout the Laurentian Channel at the time of the surveys. With regard to oxygen levels, the older water



(high NACW fraction) exhibits lower values, as more oxygen has been consumed over time since the water was ventilated from the atmosphere into the interior. In contrast, the younger LCW has only recently been ventilated and thus still carries oxygen-rich water, having just exchanged its oxygen content with atmospheric values. With the oldest water being present at
Cabot Strait, this water presumably carries the lowest amount of oxygen. This projection being consistent with findings of Blais et al. (2024), who show a sustained decline in oxygen content at Cabot Strait in 2023 and with the proxy timeseries showing a slight decrease in DO concentrations within the deep water from 2018 to 2022 (see Figure 6 – middle panel). Given the well-defined age distribution in the Gulf during 2022, future observational and modelling efforts would be valuable to directly link this distribution to DO concentrations and other key biogeochemical parameters. A simplified analysis of the
proxy time series, relating mean age and LCW fraction to oxygen concentrations, shows only weak correlations. Despite increasing mean ages and declining LCW fractions from 2018 to 2022, DO levels remained relatively stable, with only a slight decrease. This suggests that variabilities in oxygen concentrations might not be able to be directly tied to water mass changes alone and are likely influenced by additional factors.

A more comprehensive analysis involving multiple biogeochemical parameters is beyond the scope of this current study and
would require more consistent tracer surveys to additionally resolve and better capture the effects of short-term (i.e. interannual) inflow variability on the regional age distribution.

As the deep water flows from Cabot Strait towards the LSLE, the already older water in the eastern part of the Gulf will flow towards the LSLE and further increase the mean age of the deep water in the western part of the Gulf and possibly also increase hypoxia slightly. Given the transit time of around 4.7 years (Stevens et al., 2024) for the bulk of water to travel from Cabot
Strait to the LSLE, oxygen levels in the LSLE are expected to continue to drop for the next years on average (Nesbitt et al., 2025 - in revision).

Over large space and time scales, the observed change in the inflow composition may be linked to a northward shift of the Gulf Stream, which has been suggested in numerous modelling studies (e.g. Claret et al., 2018; Joyce and Zhang, 2010). This northward Gulfstream shift contracts the subpolar gyre and increases the retroflection of LCW, reducing the south-westward
transport of LCW towards the mouth of the Laurentian Channel, with implications for the supply of oxygen to the GSL (Jutras et al., 2023a).

## 6 Conclusions

Transient tracers (SF$_6$ and CFC-12) were measured for the first time in the GSL to determine ventilation timescales of the deep water on the $\sigma_\Theta = 27.26$ kg/m$^3$ isopycnal along the Laurentian Channel. These measurements not only provide evidence of an
ongoing change in deep water composition as of 2022, but also demonstrate the added value of incorporating transient tracers into water mass analysis.

The observations show a distinct pattern: near the Gulf's entrance, a signal of older, warmer, more saline NACW, while further inshore along the Laurentian Channel, younger, colder, less saline LCW is evident. This along channel gradient contrasts with



the expected age pattern based on regional estuarine circulation, which would typically predict older water closer to the LSLE. Instead, it aligns with recent studies suggesting an increasing shift towards older, warmer and saltier NACW over recent years. While previous studies have reported a 100 % NACW contribution present in the deep water in most parts of the Laurentian Channel, as of 2021, using eOMP analysis based solely on hydrographic parameters (Jutras et al., 2023b), our results show that a fraction of LCW is still present throughout the channel, though decreasing from west to east. By including tracer-derived mean ages as a parameter into a water mass analysis, we were able to capture and highlight these ongoing compositional shifts,

with a proxy time series further illustrating a temporal decrease in LCW fractions between 2018 and 2022. While the exact fraction values are subject to uncertainties, the broader trend remains evident.

The gradual, ongoing change towards increased NACW fractions may contribute to further reduction of the oxygen content within the GSL, consistent with long-term observations of Blais et al. (2024), who reported decreasing oxygen levels all throughout their time series until 2023. These findings highlight the value of including transient tracers as a parameter in future

water mass analysis to more accurately resolve temporal changes, mixing processes, and their associated biogeochemical impacts. Ultimately, the results reflect a, by 2022, continued, ongoing, measurable shift towards greater NACW influence on the GSL entering deep water.

**Appendix A: Analysis system**

The gas chromatographic – electron capture detector (GC-ECD) system consisted of a precolumn packed with 30 cm Porasil C and 60 cm Molesieve 5A, followed by a main column packed with 200 cm Carbograph 1AC and 20 cm Molsieve 5A. Throughout the measurement, these components were kept at a constant temperature of 50 °C. The function of these columns was to separate the various analytes before concentration determination with the electron capture detector.

Before the analysis, the purge and trap unit extracted the analytes from the water sample by bubbling $N_2$ gas though the sample

and trapping them on a column of 100 cm 1/16” tubing, packed with 70 cm Heysep D. To ensure efficient trapping, this column was kept at -60 to -70 °C using liquid nitrogen and subsequently heated to 100 °C to desorb the analytes onto the precolumn. The water samples were collected in 250 mL glass syringes directly from the Niskin bottles and after temporal storage in a 0 °C water bath, 200 mL of the sample volume was injected into the purge and trap unit.

The system was calibrated by measuring precise volumes of a calibrated gaseous standard containing known analyte

concentrations. A calibration curve was recorded at the beginning of each cruise and to determine any drift in the detector, point calibration was carried out daily.

Due to high variations and unusual high concentrations reported during DFO's AZMP Survey/TReX 4, we compared the results to measured samples during TReX 2 at similar density and location. This resulted in a scaling of the CFC-12 measurements by -20 % and $SF_6$ by -14 % in order to achieve internal consistency between the values (see Figure S3). We

conclude that the measured values from TReX 2 are more accurate, as they show tracer concentrations in the surface layer




close to 100 % saturation with the atmospheric values of 2022 and in general have less scatter in the data compared to TReX 4. This shift of the November cruise measurements towards plausible and similar values, and only having two datasets throughout the year limits the analysis of any interannual variability, but allows us to perform a Gulf-wide analysis of the age distribution.

**Appendix B: Transit time distribution (TTD) analysis**

The transit time distribution (TTD) is a well-established concept looking at ventilation timescales via mean age calculation using measured transient tracer concentrations. Thereby, additionally achieving knowledge on the ratio between advective and diffusive transport of water masses from the surface into the interior ocean (e.g. Stöven and Tanhua, 2014; Waugh et al., 2003). TTD determination is possible using a tracer couple sampled at the same location and time with significant different input

functions, as being the case for $SF_6$ and CFC-12 during the two cruises analyzed in this study.

The method is based on a function describing the concentration of a single tracer at a certain location ($c(t_s,r)$), calculated using boundary concentrations of this tracer, related to their input function, and Green's function ($G(t,r)$) (see Equation 5).

$$c(t_s,r) = \int_0^\infty c_0(t_s - t)e^{-\lambda t} \times G(t,r)dt \qquad (5)$$


Applying four assumptions, being (1) a steady state, (2) a single source region, (3) no inner water interactions affecting the concentration of the tracer and (4) a one-dimensional flow, and assuming an inverse Gaussian (IG) age distribution, the Inverse Gaussian Transit Time Distribution (IG-TTD) provides one solution for the TTD. Considering distinct sampling points of a tracer, Green's function ($G(t)$) at a particular time can be characterized by the mean age ($\Gamma$) and the width of the distribution

($\Delta$), excluding the location (see Equation 6) (Schneider et al., 2012; Sonnerup et al., 2013; Stöven and Tanhua, 2014; Waugh et al., 2002).

$$G(t) = \sqrt{\frac{\Gamma^3}{4\pi\Delta^2 t^3}} \times \exp\left(\frac{-\Gamma(t-\Gamma)^2}{4\Delta^2 t}\right) \qquad (6)$$

The ratio of width to mean age ($\Delta/\Gamma$), describes the above-mentioned relationship of advective to diffusive flow. A $\Delta/\Gamma$-ratio of 1.0 is the unity ratio, $\Delta/\Gamma$-ratio > 1 indicate a diffusive dominating process and a $\Delta/\Gamma$-ratio < 1 a more advective ventilation of a water parcel.

The initial attempt to constrain the TTD in this study involved comparing mean ages derived from CFC-12 to $SF_6$ measurements at the same location and time (see Figure B1a). However, this method did not yield sufficient results, as optimal

outcomes would align the data points on top or close to the linear line. One potential reason for this discrepancy could be the estimate of a steady state in ventilation, which is not the case in the Gulf (see main part of the paper). Additionally, uncertainties



may arise from the assumption of an inverse Gaussian shape for the TTD, which might not be practical to use in this area and/or the presence of only rather young water, affecting the calculations from CFC-12 measurements, given their atmospheric history of declining concentrations since 2002 (see Figure S1).

Trying to rule out this last uncertainty, we followed a different approach in taking only the $SF_6$ concentrations and calculating mean ages using varying ratios. From the resulting mean ages, the assumed CFC-12 concentrations were determined by backwards calculation, and these values were then compared to the actual measured concentrations.







**Figure B1: a) Comparison of calculated mean ages from CFC-12 and SF$_6$ with different Δ/Γ-ratio. The saturations for each tracer were assumed to be time dependent. b) Display of the relationship between calculated and measured pCFC-12 against pSF$_6$ concentrations using a ratio of Δ/Γ=1.2 for the calculations.**

Figure B1b shows the difference in the expected to measured concentrations of CFC-12 as a function of the SF$_6$ partial pressures. As previously observed in the other more classic approach, the best results are achieved when using a Δ/Γ-ratio of



> 1.4 (see Figure B1a and S4), ideally > 1.8, indicated by datapoints being closest to the yellow and blue lines. As for lower $pSF_6$ concentrations, the ratio of calculated to measured pCFC-12 is similar as to high concentrations, indicates that the influence of the decreasing pCFC-12 in the atmosphere is not very pronounced in the calculation and that the difficulty in constraining the TTD results mainly arise from the assumptions made for the calculation. Being the steady state, the single source region and the inverse Gaussian shape.

Nonetheless, since the uncertainty arises from various sources, we conclude in using a $\Delta/\Gamma$-ratio of 1.2, indicating a slightly diffusive dominated flow, along with the time dependent saturation of both tracers, analyzed by Raimondi et al. (2021) in the Labrador Sea, for the calculation of mean ages in this study. This conclusion is drawn from the results, which indicate satisfying solutions when employing a $\Delta/\Gamma$-ratio of 1.2, with negligible difference to the results obtained from using a $\Delta/\Gamma$-ratio of 1.8. Additionally, it should be noted that with increasing $\Delta/\Gamma$-ratio, the calculated mean ages become more sensitive to deviations

in saturation and tracer age (Stöven et al., 2015).

For the data inside the Gulf of St. Lawrence we primarily focus on the mean ages calculated from $SF_6$ concentrations due to rather young waters, but for the data outside the Gulf we also include data from CFC-12 concentrations due to the larger quantity of available measurements.

## Appendix C: Uncertainty estimates for mean age determination

The uncertainty in the derived mean ages arises from several factors: measurement errors in the transient tracer concentrations, uncertainties in constraining the TTD using various assumptions, and uncertainties in the input function, including atmospheric concentrations and saturation levels. The uncertainty in the transient tracer concentration involves analytical measurement errors, calibration biases ($\approx 1\,\%$), and errors arising from the use of a fixed purge efficiency in the calibration (for CFC-12: $\approx$ 2 %) (Stöven and Tanhua, 2014).

The uncertainty increases when using these tracer concentrations to determine ventilation dynamics and calculating mean ages. Consistent with previous studies, we estimate that the input function contributes a bias to the calculations with 5-10 % (DeGrandpre et al., 2006; Haine and Richards, 1995; Stöven and Tanhua, 2014; Tanhua et al., 2008). This primarily involves the use of a fixed saturation, here the time dependent saturation from the Labrador Sea (Raimondi et al., 2021), as this varies especially at higher latitudes. Additionally, it includes the assumption of non-linearity in tracer solubility, which is influenced

by mixing of different water masses in different ways. A minor contribution to the error caused by the input function is using a mean atmospheric concentration derived from measurements at various facilities in the northern hemisphere (Walker et al., 2000). Although this only introduces a bias of less than 1 %.

The assumptions applied to constrain the TTD, such as steady state conditions, a single source region, no inner water interaction, and an inverse Gaussian shape of the distribution, add another source of uncertainty to the calculation of mean

ages. The actual distribution may deviate, particularly due to the mixing of various water masses with different time histories,




as seen in the merging of the NACW and LCW. Another source of uncertainty arises from the choice of a specific Δ/Γ-ratio, which determines the balance between advective and diffusive flows.

Overall, the combined error, which includes uncertainties from the use of a fixed input function and measurement inaccuracies (the largest contributors), results in an uncertainty of up to 10 %. This error is included in the mean age results in this study.

**Appendix D: Uncertainty budget for water mass fraction analysis**

To account for uncertainties in the water mass fraction analysis, we developed a detailed uncertainty budget that explicitly incorporates individual sources of uncertainty. All sources were simultaneously propagated using a Monte Carlo simulation with 10,000 realizations.

The first source of uncertainty comes from the choice of the endmembers for each parameter, including mean age, temperature and salinity. The mean age endmembers were determined by calculating the mean of all measurements since 2010 for each water mass in the Atlantic Ocean. Therefore, the associated error was computed as the standard error of the mean (SEM), as follows:

- **NACW**
  a. Endmember range of all included measurements: 49-105 years
  b. Mean of all included measurements: 86.5 years
  c. Standard deviation: $\sigma_{\Gamma,NACW} = 13.16$
  d. $Standard\ error\ of\ the\ mean\ (SEM) = \frac{\sigma_{\Gamma,NACW}}{\sqrt{number\ of\ included\ samples}} = 1.67\ years$ (7)
  e. $Relative\ SEM = \frac{1.67\ years}{86.5\ years} = 1.93\%$ (8)

- **LCW**
  a. Endmember range of all included measurements: 7-19 years
  b. Mean of all included measurements: 12.5 years
  c. Standard deviation: $\sigma_{\Gamma,LCW} = 3.29$
  d. $Standard\ error\ of\ the\ mean\ (SEM) = \frac{\sigma_{\Gamma,LCW}}{\sqrt{number\ of\ included\ samples}} = 0.23\ years$ (9)
  e. $Relative\ SEM = \frac{0.23\ years}{86.5\ years} = 1.84\%$ (10)

For salinity and temperature endmembers, the range for each water mass were taken from Jutras et al. (2020), and the uncertainty were derived as the standard deviation of a uniform distribution over these range (see Equations 11 through 14):

- **NACW temperature**
  a. Endmember range: 4.4 °C – 8 °C (±0.2 °C)
  b. Middle of range: 6.2 °C (±0.2 °C)
  c. Standard deviation of uniform distribution: $\sigma_{\Theta,NACW} = \frac{(8\,°C - 4.4\,°C)}{\sqrt{12}} = \frac{3.6\,°C}{\sqrt{12}} = 1.04°C$ (11)



- **LCW temperature**
  a. Endmember range: -0.7 °C – 3.2 °C (±0.2 °C)
  b. Middle of range: 1.25 °C (±0.2 °C)
  c. Standard deviation of uniform distribution: $\sigma_{\Theta,LCW} = \frac{(3.2\,°C - (-0.7\,°C))}{\sqrt{12}} = \frac{3.9\,°C}{\sqrt{12}} = 1.13°C$      (12)

- **NACW salinity**
  a. Endmember range: 35 psu – 35.2 psu (±1.2 psu)
  b. Middle of range: 35.1 psu (±1.2 psu)
  c. Standard deviation of uniform distribution: $\sigma_{Sp,NACW} = \frac{(35.2\,psu - 35\,psu)}{\sqrt{12}} = \frac{0.2\,psu}{\sqrt{12}} = 0.06\,psu$      (13)

- **LCW salinity**
  a. Endmember range: 33.4 psu – 35 psu (±0.5 psu)
  b. Middle of range: 34.2 psu (±0.5 psu)
  c. Standard deviation of uniform distribution: $\sigma_{Sp,LCW} = \frac{(35\,psu - 33.4\,psu)}{\sqrt{12}} = \frac{1.6\,psu}{\sqrt{12}} = 0.46\,psu$      (14)

Furthermore, the mean ages observed within the GSL and used in the fraction analysis were assigned a relative uncertainty of 10 % (see Appendix C). To account for model simplifications associated with using a simple linear mixing model and a least square solution, an additional 20 % relative uncertainty was included.

All uncertainties were then jointly propagated using a Monte Carlo simulation with 10,000 realizations (JCGM, 2008), resulting in a final relative uncertainty of approximately 21 % for the estimated LCW fractions.

**Data availability:**

The transient tracer and CTD data of the two cruises analyzed in this study will be made available through the open research repository 'Canadian Integrated Ocean Observing System – St. Lawrence Global Observatory' (CIOOS-SLGO) via https://doi.org/10.26071/d6f3fdfc-788d-48ff. These datasets are part of an integrated data product documenting a 20-year biogeochemical time series in the Gulf of St. Lawrence region. The metadata is already publicly accessible, while the data itself will be released following the submission of a related data manuscript. Additionally, the CTD data from the DFO AZMP survey can be requested at https://open.canada.ca/en.

Historical transient tracer data used from the GLODAPv2.2022 data product can be found at https://www.ncei.noaa.gov/access/ocean-carbon-acidification-data-system/oceans/GLODAPv2_2022/.



**Author contribution:**

LG, WAN, SWS and TT conceived the study. LG, WAN, SWS and DWRW coordinated and conducted at-sea campaigns. LG conducted analysis and wrote the manuscript with writing and editorial contributions from all authors.

**Competing interests:**

The authors declare that they have no conflict of interest.

**Acknowledgments:**

We are grateful to REFORMAR and the captains and crew of the *R/V Coriolis II* for support on the cruises. We want to thank the National Research Council's Oceans program and the Department of Fisheries and Oceans for providing additional ship 560 time support for the TReX Deep experiment. Specifically, thanks to Dr. Marjolaine Blais for allowing us to participate on the 2022 AZMP cruise and for sharing the data. Finally, we would also like to acknowledge other researchers on board for their support, especially Adriana Reitano, Marshal Thrasher, and Jeshua Becker, and all scientists and technicians who contributed to the data in GLODAPv2.2022.

**Financial support:**

Financial support for the TReX project was provided by the Marine Environmental Observation, Prediction and Response (MEOPAR) Network of Centres of Excellence and the Réseau Québec maritime and its Odyssée Saint-Laurent ship time program. Partial support for student personnel and technical assistance was provided by a NSERC Discovery Grant to DWRW. SWS's participation was supported by a TReX Graduate Award, a UBC Four-Year fellowship, and a postdoctoral fellowship from the Tula Foundation.

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
