# Peer review of "The changing composition of the Gulf of St. Lawrence inflow waters observed from transient tracer measurements"

_EGUsphere, 2025_

## Referee Comment (RC1)

Review of Gerke et al.'s "The changing composition of the Gulf of St. Lawrence inflow waters observed from transient tracer measurements" (manuscript #: egusphere-2025-3999)

General comments:

The authors of this manuscript assess the mean ages of the waters in the Gulf of St Lawrence and suggest that there has been a gradual increase in the proportion of North Atlantic Central Waters from inshore areas to the entrance of the Gulf of St Lawrence is evidence of a shift towards deep waters dominated by North Atlantic Central Waters since 2022. They use transit-time distributions to derive the mean ages and then integrate them into a water mass analysis to analyze the water mass composition of the region. The authors use transient tracer measurements collected and described in Stevens et al. (2024) and use the same density surface to represent the core of the deep water inflow. CFC-12 has seen its maximum concentrations in the atmosphere, but SF6 continues to increase; CFC-12 is useful for water masses between 23-85 years in mean age whereas SF6 is useful for younger water masses and in helping to resolve the ambiguity in mean age estimates using the Inverse Gaussian transit-time distribution approach (e.g., Guo et al., 2025). Although, it seems like the authors here just relying on SF6 according to their Appendix B and back-calculate CFC-12, which makes me wonder what information the authors are getting from CFC-12. There may be data issues with regions lacking SF6 and in the regions where waters approach the time scale of CFC-12 beginning to be emitted, there are signal issues. I appreciate the trend analysis and other detailed efforts that went into this manuscript, especially to the end that the relative proportion of waters is changing, but the authors need to perform some additional analyses to convince me of their interpretation of the data beyond that. I suggest minor revisions. Specific comments are listed below:

Specific comments:

Line 36: You should add "abiotic" in front of "transient tracer concentrations" here; you could say "passive" but that would exclude radioactive transient tracers

Line 151: I'm not sure why a Delta/Gamma ratio of 1.2 was chosen; Ebser et al. (2018), which a co-author on your study was also a co-author on, found different ratios for different water masses (e.g., 0.5-0.6 where Labrador Sea Water dominates and 0.9 where North Atlantic Deep Water dominates). Please say more about Figure B1, which seems to be where 1.2 came from. Please also explain why you don't consider using a different ratio of mean age to half-width for North Atlantic Central Waters as opposed to Labrador Current Waters.

Lines 167-168: According to Guo et al. (2025), your estimates of the mean age are likely biases wherever you only use CFC-12 and no SF6 so did you see any spatial discontinuities or other signs that your estimates were different where you have SF6 vs where you do not? You can evaluate the bias you would have in regions where you have SF6 measurements by doing the mean age estimation with both CFC-12 and SF6

and again with only CFC-12 to assess the bias. Analysis was done to corroborate the CFC-12 measurements with the back-calculated CFC-12 concentrations in Appendix B where there are SF6 data but it's unclear to me what information CFC-12 is then providing.

Lines 174-177/Equations 1-4: Is the water in the Gulf of St Lawrence exclusively composed of LCW and NACW? There's also the cold intermediate layer and surface/warm slope water, I thought. Also, while the mean ages of two IG TTD for LCW and NACW would linearly sum to a new mean age, the resulting TTD will not be IG. So are you assuming that the TTDs for LCW and NACW are not IG but their sum is IG (in which case the TTDs for LCW and NACW will still need to have their means linearly combine)? Or are you going to use a sum of two IGs as your TTD?

Lines 246-247: Is the sudden discontinuity in temperature and salinity at the eastern tip of Anticosti Island physical or actually due to the availability of SF6 on one side and lack of SF6 measurements on the other?

Figures 4-5: When I see waters with mean ages of 60+ years using the tracer-based constraints you have, there becomes a signal detection issue because of the very low concentrations of CFC-12 in its first couple of decades of being emitted and the fact that you used a backwards calculation to infer the CFC-12 concentrations from the mean ages that you got from SF6 measurements (lines 450-452). Your Figure 4d makes it look like this generally is reflected in your uncertainties, but your Figure 4b has mean ages of up to 100 years, which shouldn't be detectable using CFC-12 and/or SF6. Also, your Figures 4a-b makes it look like waters are being ventilated after mixing with waters coming from the St Lawrence River in the western part of the Gulf of St Lawrence and there is a barrier for younger waters southeast of the Gulf of St Lawrence to get into the Gulf there through the Laurentian Channel, which leads to an increase in age as the waters reside for longer within the southeastern portion of the Gulf. Your interpretation is that the younger waters in the western portion of the Gulf are due to a higher portion of LCW mixing with the other waters there but is the mix of high and intermediate proportions of LCW shown in Figure 5 in the western portion of the Gulf with large variability over a small spatial distance due to potential data issues such as the ones I've pointed out in this comment and others? For example, you tend to have higher proportions of LCW where you don't look like you have SF6 measurements in the western part of the Gulf.

Figure 6: I'm not sure what the purpose of showing the relative stability of the oxygen concentrations is here because oxygen concentrations can change due to respiration changes, which isn't part of your analysis here. If you use your TTDs to calculate the preformed oxygen, on the other hand, then that may be worth showing. This figure, on the other hand, does show a trend in the variables that support your interpretation of the relative proportion of NACW vs LCW changing.

Lines 327-330: Where was this shift previously reported to be occurring, specifically?

---

## Author Comment (AC1)

Review of Gerke et al.'s "The changing composition of the Gulf of St. Lawrence inflow waters observed from transient tracer measurements" (manuscript #: egusphere-2025-3999)

**General comments:**

The authors of this manuscript assess the mean ages of the waters in the Gulf of St Lawrence and suggest that there has been a gradual increase in the proportion of North Atlantic Central Waters from inshore areas to the entrance of the Gulf of St Lawrence is evidence of a shift towards deep waters dominated by North Atlantic Central Waters since 2022. They use transit-time distributions to derive the mean ages and then integrate them into a water mass analysis to analyze the water mass composition of the region. The authors use transient tracer measurements collected and described in Stevens et al. (2024) and use the same density surface to represent the core of the deep water inflow. CFC-12 has seen its maximum concentrations in the atmosphere, but SF6 continues to increase; CFC-12 is useful for water masses between 23-85 years in mean age whereas SF6 is useful for younger water masses and in helping to resolve the ambiguity in mean age estimates using the Inverse Gaussian transit-time distribution approach (e.g., Guo et al., 2025). Although, it seems like the authors here just relying on SF6 according to their Appendix B and back-calculate CFC-12, which makes me wonder what information the authors are getting from CFC-12. There may be data issues with regions lacking SF6 and in the regions where waters approach the time scale of CFC-12 beginning to be emitted, there are signal issues. I appreciate the trend analysis and other detailed efforts that went into this manuscript, especially to the end that the relative proportion of waters is changing, but the authors need to perform some additional analyses to convince me of their interpretation of the data beyond that. I suggest minor revisions. Specific comments are listed below:

**Response:**

We thank the reviewer for their constructive comments, which we believe will help improve the manuscript. While we will respond in detail to the specific points raised, we would first like to clarify a few possible misunderstandings.

We did analyze the gradual increase in the proportion of North Atlantic Central Waters (NACW) by assessing mean ages, and we interpret this as evidence of an ongoing shift in deep water composition as of 2022. However, previous studies (Gilbert et al., 2005; Jutras et al., 2020) also observed this shift but suggested that the transition to 100% NACW had occurred prior to 2022.

For our analysis, we relied on transient tracer data (CFC-12 and SF6), as described in this study, not as in Stevens et al. (2024). Stevens et al. (2024) focused on the tracer (CF3SF5), which had been released in 2021 on the core deep water isopycnal of  $\sigma$ =27.26 kg/m3. Because all three tracers were measured

simultaneously from the same water sample, and the main objective of the cruises was to track the released tracer (i.e. to analyze the deep water spread and dispersion over time in the Gulf of St. Lawrence), the sampling strategy was centered on this isopycnal, resulting in a high density of measurements at this depth.

This said, within the Gulf we have measurements of both SF6 and CFC-12 at all locations. Only outside the Gulf, in the North Atlantic, is a lack of SF6 measurements.

**Suggested changes in manuscript:**

In the revised manuscript, we will provide a more detailed description of these points to avoid any misunderstanding regarding what was measured and where. Also, we will revisit the methods sections to ensure that the methodology is clearly outlined.

**Specific comments:**

Line 36: You should add "abiotic" in front of "transient tracer concentrations" here; you could say "passive" but that would exclude radioactive transient tracers

**Response/Suggested changes in the manuscript:**

Thanks to the reviewer for pointing this out and we will add 'abiotic' in the revised manuscript.

Line 151: I'm not sure why a Delta/Gamma ratio of 1.2 was chosen; Ebser et al. (2018), which a co-author on your study was also a co-author on, found different ratios for different water masses (e.g., 0.5-0.6 where Labrador Sea Water dominates and 0.9 where North Atlantic Deep Water dominates). Please say more about Figure B1, which seems to be where 1.2 came from. Please also explain why you don't consider using a different ratio of mean age to half-width for North Atlantic Central Waters as opposed to Labrador Current Waters.

**Response:**

We thank the reviewer for drawing our attention to Ebser et al. (2018). That study focuses on samples collected in the Eastern Tropical North Atlantic, off the coast of northwest Africa. It reports  $\Delta/\Gamma$  values of 0.5-0.6 for the Labrador Sea Water

and  $\Delta/\Gamma$ =0.9 for the North Atlantic Deep Water. However, these are different water masses than the North Atlantic Central Water (NACW) and the Labrador Current Water (LCW). The LCW represent near surface waters within the Labrador Sea and the Newfoundland shelf area, which subsequently mix with Labrador Sea Water found at depths of 1000-2000m. North Atlantic Deep Water (present around 3000m depth) is also distinct from NACW. Notably, Ebser et al. (2018) mention Atlantic Central Waters (depths above 800m) having a  $\Delta/\Gamma$ =1.0, which likely better represents NACW conditions.

In our study we select  $\Delta/\Gamma=1.2$  for the Gulf of St. Lawrence deep water and adjacent regions near its entrance for several reasons. First, we used SF6 and CFC-12 concentrations sampled and measured simultaneously at the same location, applying the assumptions outlined in the manuscript. Following a standard approach to gain information on the  $\Delta/\Gamma$  ratio, we compared the mean ages derived from the individual tracers under varying  $\Delta/\Gamma$  ratios. Agreement between the two tracer-derived mean ages (yellow lines in Figure B1a) indicates that the chosen  $\Delta/\Gamma$  reflects local advective and diffusive transport characteristics. Since perfect agreement was not achieved across all ratios, and CFC-12-based ages were generally higher than those derived from SF6, we examined whether this discrepancy could be related to the atmospheric decline of CFC-12 since 2002. To test this, we compared measured and calculated CFC-12 concentrations (again under varying  $\Delta/\Gamma$  ratios) relative to observed SF6 (e.g. Figure B1b). The calculated values were consistently slightly higher than the measured values across the full range of SF6 concentrations (2-8ppt), with no systematic trend towards higher SF6 values representing recently ventilated waters. We therefore concluded that the atmospheric decline of CFC-12 is not a relevant factor for our TTD analysis in this region.

As shown in Figure B1a, the tracer mean ages converge toward the 1:1 line as  $\Delta/\Gamma$  was increases. However, Stöven et al. (2015) concluded that ratios approaching 1.8 make the age estimates highly sensitive to tracer saturation and age deviations. Consequently, we selected  $\Delta/\Gamma$ =1.2 as an optimal balance: it reflects slightly diffusive dominated transport ( $\Delta/\Gamma$ >1) while avoiding excessive age deviations ( $\Delta/\Gamma$ >1.6).

We consider this choice consistent with conditions in LCW and NACW in the North Atlantic, where water residence times before entering the Gulf are relatively short. Unfortunately, only CFC-12 was measured in these regions, preventing us from computing local  $\Delta/\Gamma$  ratios.

To include  $\Delta/\Gamma$  values reported in Ebser et al. (2018), as suggested by the referee, we evaluated how the inferred LCW fraction would change when applying individual  $\Delta/\Gamma$ -ratios for LCW (0.5) and NACW (0.9), when computing mean ages and mean age endmembers of the water masses. This analysis yielded even higher LCW contributions than those already inferred in our study.

In addition to further assess the plausibility of our chosen ratio, we examined  $\Delta/\Gamma$  results derived from a one-dimensional Gaussian pipe model from the spreading of the CF3SF5 tracer analyzed in Stevens et al. (2024). In this model,  $\Delta/\Gamma$  is calculated from the advective ( $\Delta$ =ut) and diffusive ( $\Gamma$ =(2kt)-1/2) terms. These resulted in values for the two surveys of 1.2 and 1.5, respectively, well within the same range as the ratio applied in our study, thus providing additional support for our choice.

**Suggested changes in manuscript:**

In the revised manuscript, we intent to provide a more detailed description of Figure B1 and explain more clearly how we justify the use of  $\Delta/\Gamma$ =1.2 across all regions, both within the Gulf of St. Lawrence and for LCW and NACW. This description will follow the response outlined here. In addition, we will add a note on the uncertainty of mean ages outside the Gulf, where only CFC-12 measurements are available, as discussed by Guo et al. (2025) (see comment below).

Lines 167-168: According to Guo et al. (2025), your estimates of the mean age are likely biases wherever you only use CFC-12 and no SF6 so did you see any spatial discontinuities or other signs that your estimates were different where you have SF6 vs where you do not? You can evaluate the bias you would have in regions where you have SF6 measurements by doing the mean age estimation with both CFC-12 and SF6 and again with only CFC-12 to assess the bias. Analysis was done to corroborate the CFC-12 measurements with the back-calculated CFC-12 concentrations in Appendix B where there are SF6 data but it's unclear to me what information CFC-12 is then providing.

**Response:**

We thank the reviewer for raising this point. As explained in our response to the previous comment, the simultaneous sampling of CFC-12 and SF $_6$  throughout the Gulf ensures that  $\Delta/\Gamma$  can be chosen consistently, and that we have both tracers sampled at all locations. Thus, there are no discontinuities in mean age estimates between sites with and without SF $_6$  measurements in this region.

In the adjacent Atlantic, SF6 measurements are indeed lacking, and as Guo et al. (2025) have shown, mean ages based solely on CFC-12 can be biased when applying a fixed  $\Delta/\Gamma$  without local determination. In our analysis, however, we use the Atlantic data only to calculate multi-year average mean ages, where the associated uncertainty in the averages is weighted more heavily than the exact choice of  $\Delta/\Gamma$ .

**Suggested changes in manuscript:**

In the revised manuscript we intent to describe the potential bias in the mean age estimates for the Atlantic Ocean, as they are only based on CFC-12 measurements, and we will refer to Guo et al. (2025).

Lines 174-177/Equations 1-4: Is the water in the Gulf of St Lawrence exclusively composed of LCW and NACW? There's also the cold intermediate layer and surface/warm slope water, I thought. Also, while the mean ages of two IG TTD for LCW and NACW would linearly sum to a new mean age, the resulting TTD will not be IG. So are you assuming that the TTDs for LCW and NACW are not IG but their sum is IG (in which case the TTDs for LCW and NACW will still need to have their means linearly combine)? Or are you going to use a sum of two IGs as your TTD?

**Response:**

We thank the reviewer for this comment. The deep water we focus on in this study does consists solely of LCW and NACW. While intermediate and surface waters are present in the Gulf, they do not occur at the  $\sigma$ =27.26 kg/m³ isopycnal. At this depth mixing with surface and intermediate waters is highly unlikely, as also present in Jutras et al. (2020) 'While the contribution of the CIL is important for the intermediate waters of the Laurentian Channel (100–150 m depth), the deep waters (below 150 m) are composed almost exclusively of a mixture of LCW and NACW (see Section 3.1).'.

We appreciate the reviewer raised the point about the combination of two IGTTDs. As shown by Stöven and Tanhua (2014), a 2-IG-TTD approach can be applied, in which mean ages from two different water masses are linearly combined using a mixing factor  $\alpha$  (see the following equation).

$$\Gamma = \alpha * \Gamma_1 + (1 - \alpha) * \Gamma_2$$

For our case, assuming  $\Delta/\Gamma=1.2$  for both NACW and LCW, the average mean ages are  $\Gamma1=86.5$  years and  $\Gamma2=12.5$  years, respectively. The mixing factor  $\alpha$  can then be determined for each computed mean age value within the Gulf. The analysis

yields results, that are consistent with those obtained from our water mass analysis based on temperature, salinity and mean age (see Figure below).

Figure: Distribution factor  $\alpha$  on the  $\sigma_0$  = 27.26 kg/m³ isopycnal deep water in the Laurentian Channel from the 2-IG-TTD analysis using observed mean age ( $\Gamma$ ) plotted on a map of the Gulf of St. Lawrence.

**Suggested changes in manuscript:**

In the revised manuscript, we will briefly include the results of the 2-IG-TTD analysis. We do not intent to go into detail on the method itself, as this would considerably extend the method section, but instead refer to Stöven and Tanhua (2014). We will also highlight that the results are consistent with the water mass fraction analysis based on three parameters.

Lines 246-247: Is the sudden discontinuity in temperature and salinity at the eastern tip of Anticosti Island physical or actually due to the availability of SF6 on one side and lack of SF6 measurements on the other?

**Response:**

SF6 and CFC-12 were measured simultaneously at all locations within the Gulf of St. Lawrence. Therefore, the observed discontinuity in temperature and salinity

near the eastern tip of Anticosti Island is not related to tracer availability and is most likely of physical origin.

**Suggested changes in manuscript:**

We do not plan to add specific information on this point in the revised manuscript. However, by improving the overall description of tracer usage and sampling in the method section, we aim to avoid such misunderstanding.

Figures 4-5: When I see waters with mean ages of 60+ years using the tracer-based constraints you have, there becomes a signal detection issue because of the very low concentrations of CFC-12 in its first couple of decades of being emitted and the fact that you used a backwards calculation to infer the CFC-12 concentrations from the mean ages that you got from SF6 measurements (lines 450-452). Your Figure 4d makes it look like this generally is reflected in your uncertainties, but your Figure 4b has mean ages of up to 100 years, which shouldn't be detectable using CFC-12 and/or SF6. Also, your Figures 4a-b makes it look like waters are being ventilated after mixing with waters coming from the St Lawrence River in the western part of the Gulf of St Lawrence and there is a barrier for younger waters southeast of the Gulf of St Lawrence to get into the Gulf there through the Laurentian Channel, which leads to an increase in age as the waters reside for longer within the southeastern portion of the Gulf. Your interpretation is that the younger waters in the western portion of the Gulf are due to a higher portion of LCW mixing with the other waters there but is the mix of high and intermediate proportions of LCW shown in Figure 5 in the western portion of the Gulf with large variability over a small spatial distance due to potential data issues such as the ones I've pointed out in this comment and others? For example, you tend to have higher proportions of LCW where you don't look like you have SF6 measurements in the western part of the Gulf.

**Response:**

We thank the reviewer for pointing this out, and we agree that more explanation is necessary. Again,  $SF_6$  and CFC-12 were measured simultaneously at all locations within the Gulf of St. Lawrence, and the backward-calculated CFC-12 concentrations were only used for the TTD analysis, not for any direct data evaluation.

The fact that mean ages exceed the atmospheric age of a tracer (e.g., 85 years for CFC-12) arises from the interpretation through the transit time distribution (TTD). Mean ages do not simply reflect the time since a water parcel last contacted the atmosphere (tracer age), when analyzing the tracer concentration. Instead, it accounts for the distribution and mixing of water masses. The tail of the TTD

represents older water, so the mean age can exceed the tracer's atmospheric lifetime, especially when tracer concentrations are low. As noted by Guo et al. (2025) as well, 'for water with an ideal age under 200 years, the CFC-12-based IGTTD can provide meaningful mean ages up to this limit, despite the tracer's shorter ~80-year atmospheric history.'

The deep waters at  $\sigma$ =27.26 kg/m³ are separated from surface waters by the cold intermediate layer, so young waters from the St. Lawrence River are unlikely to influence them. The increase in mean age towards the St. Lawrence Estuary is instead due to a higher fraction of LCW. As our analysis focuses on data east of the Lower St. Lawrence Estuary, before any upwelling of deep water towards the surface occurs, influence of surface St. Lawrence River seems unlikely at these depths (As also stated by Jutras et al. (2020), as shown in the previous comment). Given the general circulation pattern of deep water into the Gulf and surface water out, the observed mean age variability primarily reflects processes within the deep water of the Laurentian Channel.

Following a comment from M. Jutras, we also examined other isopycnals along the Laurentian Channel in the deep water and observed some mixing with younger LCW from shallower deep water regions. However, data coverage is limited, as our sampling targeted  $\sigma_0 = 27.26 \text{ kg/m}^3$ .

**Suggested changes in manuscript:**

In the revised manuscript, we intent to add a figure of mean ages and water mass fractions throughout the deep water layer, plotted against distance to Cabot Strait. This analysis will illustrate the internal mixing during the transit through the Laurentian Channel. We will also clarify that the influence of the St. Lawrence River on these deep waters is unlikely.

Figure 6: I'm not sure what the purpose of showing the relative stability of the oxygen concentrations is here because oxygen concentrations can change due to respiration changes, which isn't part of your analysis here. If you use your TTDs to calculate the preformed oxygen, on the other hand, then that may be worth showing. This figure, on the other hand, does show a trend in the variables that support your interpretation of the relative proportion of NACW vs LCW changing.

**Response:**

We thank the reviewer for this comment. The respiration rate was analyzed in detail in Nesbitt et al. (2025) and has been considered in our analysis. We

included the oxygen measurements primarily to illustrate that using oxygen to calculate water mass fractions is challenging, and to highlight in the discussion that oxygen concentrations still show a slight decrease, consistent with Blais et al. (2024).

**Suggested changes in manuscript:**

In the revised manuscript we plan to move the oxygen plot to the Supporting Information, leaving only the mean age and LCW fraction time series figure. This will still allow us to refer to the plot to make the two relevant points mentioned in the response. Additionally, we will add that the number of data points does not allow to refer this to a statistically significant trend, following an analysis pointed to by a comment from M. Jutras.

Lines 327-330: Where was this shift previously reported to be occurring, specifically?

**Response:**

We thank the reviewer for this comment. Previous studies report this shift for the deep water of the Gulf of St. Lawrence, specifically at the Cabot Strait and throughout the Laurentian Channel.

**Suggested changes in manuscript:**

In the revised manuscript, we intent to add this information on specific locations to each referenced historic fraction analysis.

---

## Author Comment (AC2)

Comment to manuscript "The changing composition of the Gulf of St. Lawrence inflow waters observed from transient tracer measurements" by Gerke et al.

By M. Jutras

Gerke et al provide here a new and very useful estimate of water age distribution in the St. Lawrence System. This is the first assessment based on transient tracers in this region, and it helps answer important questions of the circulation and how it is changing.

1. First of all, I have a question regarding the role of vertical mixing along the Laurentian Channel. On Fig. 4C, right panel, we see that the age is strongly dependent on the density, with older (hence NACW-rich) waters at depth. In Fig. 5 of Jutras et al. (2020), the results from the eOMP suggest that LCW and NACW are not mixed when they enter the Laurentian Channel, and that they mix as they move inland. More specifically, LCW progressively mixes with NACW on the 27.25 isopycnal, which would result in waters getting younger as they move inland along that isopycnal. Therefore, I would suggest to show the results of the water mass analysis also for different densities, as was done for the rest of the results (with figures like 4c and 3a-d, or maybe using a vertical transect along the channel), since this might influence the interpretation of the results that the rejuvenation of waters as they move inland is due to temporal changes in the water mass composition.

**Response:**

We thank M. Jutras for this insightful suggestions regarding other density layers along the Laurentian Channel. Our study focused on the  $\sigma_0$  = 27.26 kg/m³ isopycnal because the sampling strategy for transient tracers targeted this deep water layer.

However, by analyzing mean ages and computed water mass fractions at other densities within the deep water, we observe that older water resides at depth while younger water is found closer to the Cold Intermediate Layer (CIL). This vertical structure appears to influence the 27.26 isopycnal, with water ages increasing along the Laurentian Channel transport inland (see Figure).

Despite this, LCW remains present throughout the deep water of the Laurentian Channel in 2022, while pure NACW is only found at the bottom near Cabot Strait.

Figure: Display of different variables throughout the Laurentian Channel (shown as distance from Cabot Strait, where greater distance corresponds to locations farther toward the Lower St. Lawrence Estuary). From top to bottom: Temperature, Salinity, Mean Age and LCW fraction within the deep water ( $\sigma_{\Theta} > 26.25 \text{kg/m}^3$ ), with the shaded area representing the density surface of  $\sigma_{\Theta} = 27.26 \text{kg/m}^3$ .

**Suggested changes in manuscript:**

In the revised manuscript, we intent to add this or a similar plot plot along with a brief description of the analysis, noting that vertical mixing also plays a role. Nevertheless, the influence of LCW remains evident throughout the deep water of the Laurentian Channel.

2. Second, I was wondering if the effect of turbulent mixing on SF6 and CFC-12 concentrations had been estimated, and if there could be some contamination by surface waters that would make the waters younger as they move inland. This effect is mentioned in the discussion, but the magnitude is not quantified.

**Response:**

Turbulent mixing was not examined and accounted for in the calculations. However, its effect is expected to be minor because the cold intermediate layer effectively separates the deep water from the younger surface waters. This was also stated in Jutras et al., 2020: "While the contribution of the CIL is important for the intermediate waters of the Laurentian Channel (100–150 m depth), the

deep waters (below 150 m) are composed almost exclusively of a mixture of LCW and NACW (see Section 3.1)". Additionally, our analysis focuses on data within the Laurentian Channel and does not extend into the Lower Estuary. Mixing of deep water towards the surface along the continental shelf would primarily affect surface waters and is unlikely to significantly alter the age of the deep water we analyze. This upwelling also primarily takes place at longitudes of the Lower St. Lawrence Estuary, and we only consider data east of this region.

**Suggested changes in manuscript:**

We do not intent to change the analysis for this point, but we will clarify this in the manuscript, mentioning that surface water is unlikely to effect the deep water at this isopycnal and depth.

3. Just a thought, but could the  $\theta$  and Sp water mass analysis be used to "correct" for changes in age due to changes in water mass composition, and then use the age estimates to deduce the advection time along the Laurentian Channel?

**Response:**

The temperature and salinity are incorporated in the mean age calculations, as they are included in the analysis of the partial pressure of SF6 and CFC-12. Therefore, changes in temperature and salinity are already included in the mean age used for the water mass composition calculation.

**Suggested changes in manuscript:**

Nothing to add/change within the revised manuscript.

**Specific comments:**

4. It is not clear what information the zooms on narrow density ranges provide in Fig. 3c-d and 4d. They might not be necessary.

**Response:**

The zooms on narrow density ranges are intended to highlight the  $\sigma_{\Theta}$  = 27.26 kg/m³ isopycnal, which was the main sampling focus for both tracers during the campaigns. This density surface provides the lowest uncertainty due to the number of measurements, and forms the basis for the water mass analysis

presented later. The values included in the water mass analysis are the ones shown in these 3e/f and 4a.

**Suggested changes in manuscript:**

No changes are planned in response to this comment, as the zoomed-in plots are essential for emphasizing the key isopycnal and supporting the subsequent analysis from our point of view.

5. In Fig. 6b, it would be relevant to mention if there are enough data points to confirm if the trend is statistically significant, using a statistical test. It looks like the trend would change significantly if for instance the last point was removed, which suggests that the trend is not statistically significant. There is significant lateral variability in water properties across Cabot Strait due to the circulation (as we can see on some of the maps), which could contribute to the variability observed here. Colouring the data points with longitude or distance from one end of the channel could allow to see if this explains part of the variability instead of temporal variability.

**Response:**

Thank you for raising this point. We assessed the slightly negative slope in the linear analysis and found it is not statistically significant. The 95% confidence interval of the slope includes zero, indicating the trend could be absent or even slightly positive. Furthermore, a sample size analysis using Fisher's z-transform (a method to asses the statistical significance of correlations) indicates that detecting a trend of this magnitude with 80% statistical power would require approximately 500 datapoints, whereas our analysis only contains 45.

Color-coding the data points by longitude, as suggested, is a good idea. Although just based on the calculation, the values present in 2018 come from datapoints furthest away from Cabot Strait. Nonetheless, we attach the plot including longitude as a color code here.

Figure: A proxy timeseries of dissolved oxygen at Cabot Strait from 2018 to 2022. The color code shows the actual longitude of the measurement, with samples located further west in blue and closer to Cabot Strait in red.

**Suggested changes in manuscript:**

In the revised manuscript we intent to move the oxygen plot to the supporting information and include a brief mention of the statistical analysis there. In the main text, we will add the arising uncertainty, when referring to this possible slight decrease in the oxygen over time.

---

## Author Comment (AC3)

**Dear authors,**

Please find below my review or your manuscript entitled "The changing composition of the Gulf of St. Lawrence inflow waters observed from transient tracer measurements". This manuscript describes the composition of the waters of the Gulf of St Lawrence (GSL) using new measured transient tracers. While the science is sound, and the writing relatively good, I wonder what the novelty of this science contributions is. Some results are completely expected and not new, while other puzzling results (gradient of the mean age along the Laurentian Channel) have not been fully exploited in my view. Overall, I am afraid that this study would bring more confusion than clarity in the literature of the GSL. I recommend, if not a rejection, an in-depth re-structuration of this study. I provide below some overall comments and then comments for each section.

**General response:**

We thank the reviewer for their constructive assessment and feedback. We fully acknowledge the need for clearer structure and stronger emphasis on the study's novelty. In response, we will undertake a substantial revision of the manuscript, including:

- Reorganizing the discussion section to highlight the new insights provided by transient tracer measurements and their implications for ongoing changes in the Gulf of St. Lawrence ventilation.
- Integrating a full deep-water analysis to complement the isopycnal-based results and better illustrate vertical mixing processes of the deep water along the Laurentian Channel.
- Restructuring the manuscript to improve clarity and focus, ensuring that results and interpretations are not to question established understanding of the general estuarine circulation.

We hope that these revisions will address the reviewer's concerns about novelty and clarity, and demonstrate the scientific value of the new transient tracer observations in advancing understanding of GSL deep water ventilation.

Below, we address the reviewer's specific comments in detail.

**OVERALL COMMENTS**

One of the key results of this study seems to call into question decades of research in the GSL and our basic understanding of the estuarine circulation, that is that the water is getting rejuvenated as it travels from the ocean to the head of the estuary. E.g. in the Discussion: "The temperature, salinity and mean age analysis of the deep water on the

 $\sigma\Theta$  = 27.26 kg/m3 isopycnal along the Laurentian channel show young, cooler, and fresher water in the Lower Estuary and western Gulf, with older, warmer and more saline water entering through Cabot Strait and within the eastern Laurentian Channel (Figures 3 and 4a). These are the opposite pattern of water mass age that might be expected if water is transported along the Laurentian Channel..."

I think that one problem with this study may be that the analysis is done over a single isopycnal. Is it possible that this isopycnal shoals as the water is advected toward the head of the estuary, giving the impression that the water is getting younger (being enriched by vertical mixing)? What about the same analysis over the bottom waters?

My second concern is that decadal cycles are completely ignored. The study is limited to 2018-2022. How those years related to ~100 years of observations in the GSL? Do we know anything about decadal changes in the circulation?

Finally, I am unsure what is the novelty in this study. I finished the reading thinking "so what?" If the authors decide to revise this study, I recommend making clear in which aspect this study contributes to new knowledge about the GSL.

**Response:**

We thank the reviewer for their constructive comments, and we appreciate that the novelty of the work was not sufficiently clear in the initial version. From our perspective, the main novelties of this study extending GSL research are:

- 1. The observed ongoing shift towards increased NACW contributions to the deep water, as of 2022, which had been observed to have fully happen prior to our analysis.
- 2. The use of transient tracers in water mass analysis, to include insights on circulation and mixing processes.

We do not intend to question decades of established understanding of the GSL estuarine circulation. The statement quoted by the reviewer does not suggest that the water becomes rejuvenated within the Laurentian Channel. It rather highlights that the relative contribution of NACW, before entering the channel, has increased over time. Consequently, the western part of the system shows higher proportions of LCW, resulting in younger, cooler, and fresher deep water. The misunderstanding may stem from the concept of 'age' as we have used it in this study. We use ventilation age as a water mass tracer, rather than as a measure of residence time in the estuary.

We acknowledge the reviewer's point regarding the limitation of focusing on a single isopycnal. This focus was due to the sampling strategy, which provided the highest density of data at  $\sigma_0 = 27.26$  kg/m3. However, we recognize that this

approach constrains our ability to assess vertical mixing along the deep-water inflow from Cabot Strait to the Lower St. Lawrence Estuary. In the revised version, we plan to expand the analysis to include the full deep-water section, which will allow us to better illustrate both vertical mixing and the limited shoaling of the isopycnal towards the estuary head (see Figure). However, there are limited datapoints throughout the deep water.

Regarding decadal variability, we fully agree that it represents an important aspect of GSL dynamics. Unfortunately, as this study presents the first transient tracer measurements in the region, the available dataset is temporally limited to the year of 2022. While we cannot address decadal cycles directly, we hope that this work highlights the value of including transient tracers in future long-term monitoring efforts, which could eventually enable such analyses.

Figure: Display of various variables along the Laurentian Channel deep waters as a function of depth (shown as distance from Cabot Strait, where greater distance corresponds to locations farther toward the Lower St. Lawrence Estuary). From top to bottom: Temperature, Salinity, Mean Age and LCW fraction. The black line indicates the  $\sigma_{\Theta}$ =27.26kg/m³ isopycnal for all measurements at this isopycnal included in the analysis. The grey lines represent the 26.5, 26.75, 27.0, and 27.5 isopycnals. (As a reference, the Lower St. Lawrence Estuary starts at a distance of about 600km from Cabot Strait)

**Suggested changes in manuscript:**

In the revised manuscript we intent to:

Clarify the novel contributions of this study.

- Expand the analysis to a full deep-water section, showing that the  $\sigma_0$  = 27.26 kg/m³ isopycnal does not significantly shoal, but include vertical mixing processes along the Laurentian Channel. We intent to add this or a similar figure to the manuscript.
- Emphasize that transient tracer measurements are of high value, when analyzing water masses, including mixing processes.
- Clarify at the beginning of the manuscript that the reported mean ages refer to ventilation ages, used as a water mass tracer, and do not represent residence time within the estuary.

**1. INTRODUCTION**

The Introduction is well written. I don't have many comments except better explaining the SF6 and CFC-12, their relevance, and why it is useful to measure them (how it works).

**Response:**

We thank the reviewer for this helpful comment. We agree that a clearer explanation of the relevance and application of SF6 and CFC-12 will improve the introduction.

**Suggested changes in manuscript:**

We intend to expand the first paragraph and better describe how these transient tracers are used in ocean ventilation studies and why they provide valuable insights into water mass age, supported by relevant literature references.

**2. HYDROGRAPHY**

Well written.

**3. 2 OBSERVATIONS**

- I would like more information about the transient tracers. You need to specify that this is not only valid for the GSL, but for the global ocean as well (e.g. first paragraph).

- GLODAP database is mentioned. Can you say more about this database and how representative it is? How the database was interrogated, etc. I am specifically thinking of figure 4b: why only those points are on the map? I am sure much more of Atlantic Canada was sampled.

**Response/ Suggested changes in manuscript:**

We thank the reviewer for these valuable comments. We agree that additional clarification is needed regarding the global context of transient tracers and our use of the GLODAP database. In the revised manuscript, we will aim to give a broader explanation on the relevance of transient tracers such as  $SF_6$  and CFC-12 for the world oceans.

Regarding the GLODAP database, we acknowledge that much more data are available for the North Atlantic and Labrador Sea than what is shown in Figure 4b. For our analysis, we specifically selected stations near the entrance of the GSL to ensure consistency in the choice of the  $\Delta/\Gamma$ -ratio and to obtain the closest possible reference measurements for defining the boundary conditions of the LCW and NACW mean ages. We will add more details to clarify the data selection process and explain why only those stations are displayed and used.

**3.4 CABOT STRAIT**

- "A fixed location within Cabot Strait (47.2 °N; 59.7 °W) was selected, and the distance from each sampling point to this location was calculated."
- -> How this works? The exact same station was sampled all the time? Figure 6 suggests that several dozens of points are from Cabot Strait between 2018 and 2022. These are all from the exact same location?

**Response:**

We thank the reviewer for pointing out this potential confusion. We did not sample directly at the fixed Cabot Strait location. Instead, our goal was to construct a time series representative of the Cabot Strait inflow, based on measurements collected in 2022. Using the transit time estimates from Stevens et al. (2024) and the known sampling positions along the Laurentian Channel, we inferred the approximate time when each sampled water parcel would have

passed through Cabot Stait. Consequently, data collected farther west in the Laurentian Channel were assigned earlier equivalent years (e.g. 2018) when they passed through Cabot Stait, reflecting the estimated four-year transit time through the Laurentian Channel.

**Suggested changes in manuscript:**

We will carefully review and revise the corresponding text to clearly explain that the Cabot Strait time series is a reconstructed dataset, derived from 2022 measurements combined with estimated water parcel transit times

- "Using a transit speed of 0.5 cm/s..."
- -> It was argued in section 2 that several estimations of the speed have been done. Why this one more than another?

**Response:**

We thank the reviewer for this comment. We selected the transit speed of 0.5 cm s-1 because it was determined from the tracer release experiment described in Stevens et al. (2024). This experiment measured the dispersion from released CF3SF5, a tracer measured simultaneously to the transient tracers analyzed in our study. Therefore, this estimate provides the most consistent and directly comparable transit speed for the conditions and tracers used here.

- "To account for changes in DO concentrations, the oxygen utilization rate (OUR) within the Laurentian Channel, estimated to be 21.1  $\mu$ mol/kg per year (Nesbitt et al., 2025 in revision), was considered in the calculation."
- -> As far as I can tell, OUR is not used further in the text. I am unsure why this sentence is here.

**Response:**

We thank the reviewer for this observation and this was raised by other reviewers as well. The use of OUR was indeed part of the time series estimation for Cabot Strait, where we applied the value of 21.1  $\mu$ mol/kg per year to the back-calculated oxygen values at the Cabot Strait inflow. We acknowledge that this was not clearly described in the text, which may have caused confusion.

**Suggested changes in manuscript:**

We will revise the manuscript to clarify, how the OUR was used in estimating the Cabot Strait time series. In addition, we plan to move most of the oxygen-related data and figures to the supporting information, as oxygen is not a primary focus of this study.

**4. RESULTS**

- Most of the sub-sections here are very thin... Section 4.4, for example is 2 sentences! Overall, it is not clear to me what is novel here.

**Response/Suggested changes in manuscript:**

We thank the reviewer for their helpful comment and agree that some subsections are too thin. In the revised manuscript, we will combine short sections and expand on them where appropriate to provide a more detailed explanation of the results.

- Figures 3 to 6: recall dates/time presented in this figure and where the data come from (TREX or GLODAP?). From one figure to the other, it looks like different data source are used but this information does not appear in the caption.

**Response/Suggested changes in manuscript:**

We thank the reviewer for this useful point of view, that the figure captions are missing information. We agree that the data sources and time frames should be stated more clearly and in the revised manuscript, we will add detailed descriptions to each figure legend. Specifying their data origin (TreX or GLODAP) and sampling years. (Except for figure 4b, all data is coming from the TreX missions in 2022.)

- section 4.2: "we observe oldest water (70 105 years) to be located south and west of Cabot Strait with considerably younger water located to the northeast (5 20 years)."
- -> Can you name those places? I am assuming that you refer to the Scotian shelf and the Newfoundland shelf near Flemish Cap.

**Response/Suggested changes in manuscript:**

We thank the reviewer for highlighting the need for a more detailed geographical description. We will add this in the revised manuscript by specifying that the NACW is primarily represented by waters over the Scotian shelf and southeast of the Scotian shelf, while the LCW corresponds to waters near the Newfoundland Shelf and Flemish cup.

- L.260: "Consistent with the temperature and salinity spatial variability, deep layer mean age shows an abrupt 5 10 year shift towards younger waters at approximately 63 °W (i.e., the southeastern tip of Anticosti Island; see Figure 4c)."
- -> First, there is no longitude in Figure 4c; Second, I do not understand where to look for this "shift"

**Response/Suggested changes in manuscript:**

We thank the reviewer for this observation. You are correct that the longitude is not directly visible in Figure 4c. The difference in mean age is represented by color-coding data points, where blue symbols correspond to stations west of 63°W and red symbols to those east of it. We will clarify this in the figure legend and revise the manuscript text to explain more clearly where and how this shift is observed.

- Section 4.3: I think that it is worth recalling the type of analysis here. "water mass analysis" is broad.

**Response/Suggested changes in manuscript:**

In the revised manuscript we will briefly restate the type of analysis performed in this section.

- L. 287: "Despite inherent uncertainties that limit the interpretation of the estimated fraction values, observational evidence suggests that the fraction of LCW gradually decreases from the west to the east within the Laurentian Channel."
- -> Figure 5 suggests that there is 100% LCW in the Lower estuary. Is this realistic? If true, would it mean that we have been wrong in our understanding of the system for several decades?

**Response/Suggested changes in manuscript:**

The thank the reviewer raising this important point. The apparent 100% LCW fraction near the Lower St. Larence Estuary indeed results from the uncertainties inherent in the water mass fraction calculations, but it should demonstrate that LCW is still present within the Gulf of St. Lawrence. Our findings do not contradict the established understanding of the Laurentian and Estuarine circulation pattern. Instead, they suggest that the transition towards a greater NACW contribution, previously predicted to have fully reached dominance, is still ongoing as of 2022. We will add this information more detailed to the text and figure in the revised manuscript.

**5. DISCUSSION**

- L. 305: "The temperature, salinity and mean age analysis of the deep water on the  $\sigma\Theta$  = 27.26 kg/m3 isopycnal along the Laurentian channel show young, cooler, and fresher water in the Lower Estuary and western Gulf, with older, warmer and more saline water entering through Cabot Strait and within the eastern Laurentian Channel (Figures 3 and 4a). These are the opposite pattern of water mass age that might be expected if water is transported along the Laurentian Channel..."
- -> This is a strong statement that bring confusion. See my overall comments.

**Response/Suggested changes in manuscript:**

We thank the reviewer for this important comment and agree that the original phrasing could create confusion. Our intention is not to challenge the established understanding of the Gulf of St. Lawrence and Lower St. Lawrence Estuary circulation pattern. Rather, our results indicate that the transition towards increased NACW contribution is still ongoing as of 2022, and has not yet reached full dominance through the deep waters of the Laurentian Channel.

To avoid misinterpretation and any misunderstanding, we will revise that statement cited by the reviewer in combination with the statement listed in the next comment. In doing so, we will aim to clearly outline that this study does not question any general estuarine circulation pattern and emphasize the novel aspect of this study, particularly the use of transient tracers to analyze ventilation patterns and water mass contributions, sampled for the first time in the Gulf of St. Lawrence. This was pointed out by the reviewer to include more in the discussion.

- L.314: "These observations of water mass ages support the general hypothesis of a recent change in the composition of the Gulf of St. Lawrence's deep inflow towards an increased NACW composition"
- -> Does it? two sentences before you say that these observations are opposite to what is commonly known... Again, this adds more confusion than answers.

**Response/Suggested changes in manuscript:**

We thank the reviewer for pointing out this potential inconsistency and for highlighting where our wording may lead to misunderstanding. Our observations indeed support the general hypothesis that the deep water inflow to the Gulf of St. Lawrence is shifting toward greater NACW contribution. However, our results also indicate that this transition is still ongoing rather than fully complete, which constitutes the key novelty of this study. We will revise the discussion to clarify this distinction and ensure that the interpretation of our findings is consistent throughout the text.

- L. 320 to 338: Nothing new is learn here, I don't think that this belongs in the Discussion.

**Response/Suggested changes in manuscript:**

We agree with the reviewer that this section does not contribute new insights to the discussion. We will therefore remove the paragraph summarizing previous studies and their NACW contribution estimates to maintain focus on the novel findings of this work. However, we will include them in the introduction to provide context and link our results to previous research.

- L. 350: "The oldest water with the lowest LCW fraction in 2022 is observed all the way near Cabot Strait, indicating that fractions of LCW were present throughout the Laurentian Channel at the time of the surveys."
- -> I am not sure I understand what this means.

**Response/Suggested changes in manuscript:**

The intended meaning is that the lowest LCW fractions were observed near Cabot Strait, while higher LCW contributions appeared farther west within the Laurentian Channel. This pattern suggests that the LCW contribution decreases over time as the water flows from Cabot Strait toward the Lower St. Lawrence Estuary. Even when including a look of the entire deep water column, which adds vertical mixing of fresher and younger water, this shift is still visible and adds to

the general novelty of the study, that LCW contributions are present within the Laurentian deep channel waters in 2022. We will revise this sentence for clarity in the manuscript to ensure the interpretation is unambiguous.

- L. 351- 354: again, the fact that the oxygen is consumed as the water travels in the estuary is nothing new.

**Response/Suggested changes in manuscript:**

We thank the reviewer for this comment and agree that the observation of oxygen and oxygen consumption along the estuarine pathway is not focus of this manuscript. Our intention in including this variable to the analysis was to place our results within the broader context of the known oxygen decline in the GSL, thereby demonstrating consistency with the established understanding and previous findings. We will rephrase this part of the text to make this purpose clearer, however, moving most parts of the oxygen analysis to the supporting information and referring to that.

**6. CONCLUSION**

- The first sentence recalls the novelty of this study. More of these statements should appear in the Discussion as well.

**Response/Suggested changes in manuscript:**

We thank the reviewer pointing to this sentence and we agree that the discussion would benefit from a stronger emphasis on the studies novel aspects. In the revised manuscript, for which we will integrate clearer statements throughout the discussion highlighting the first sampling of transient tracers and the ongoing shift towards higher NACW contributions, as already outlined in the response to earlier comments.